# A spin-crossover framework endowed with pore-adjustable behavior by slow structural dynamics

Jin-Peng Xue[1], Yang Hu[1], Bo Zhao[1], Zhi-Kun Liu[1], Jing Xie [1], Zi-Shuo Yao [1✉] & Jun Tao [1✉]

Host-guest interactions play critical roles in achieving switchable structures and functionalities in porous materials, but design and control remain challenging. Here, we report a two-dimensional porous magnetic compound, $[Fe^{II}(prentrz)_2Pd^{II}(CN)_4]$ (prentrz = (1$E$,2$E$)−3-phenyl-N-(4H-1,2,4-triazol-4-yl)prop-2-en-1-imine), which exhibits an atypical pore transformation that directly entangles with a spin state transition in response to water adsorption. In this material, the adsorption-induced, non-uniform pedal motion of the axial prentrz ligands and the crumpling/unfolding of the layer structure actuate a reversible narrow quasi-discrete pore (nqp) to large channel-type pore (lcp) change that leads to a pore rearrangement associated with simultaneous pore opening and closing. The unusual pore transformation results in programmable adsorption in which the lcp structure type must be achieved first by the long-time exposure of the nqp structure type in a steam-saturated atmosphere to accomplish the gate-opening adsorption. The structural transformation is accompanied by a variation in the spin-crossover (SCO) property of $Fe^{II}$, i.e., two-step SCO with a large plateau for the lcp phase and two-step SCO with no plateau for the nqp phase. The unusual adsorption-induced pore rearrangement and the related SCO property offer a way to design and control the pore structure and physical properties of dynamic frameworks.

[1] Key Laboratory of Cluster Science of Ministry of Education, School of Chemistry and Chemical Engineering, Beijing Institute of Technology, Beijing 102488, People's Republic of China. ✉email: zishuoyao@bit.edu.cn; taojun@bit.edu.cn

The development of flexible metal-organic frameworks (MOFs) or soft porous crystals (SPCs) whose porous structures can be reversibly altered in response to the sorption of guest molecules has attracted burgeoning attentions[1–5]. Such adaptive structural transformability, usually ascribed to the interplay between the guest molecules and flexible host frameworks, is associated with advanced properties in the field of gas separation, sensing, and magnetic switching[6–10]. In the past decades, many porous crystals that demonstrate guest-controllable porous expansion/contraction have been reported[11–26]. Two remarkable compounds are [Cu(aip)($H_2O$)](solvent)$_n$ and [Al(OH)$_x$(solvent)$_y$(TzDB)$_z$], (aip = 5-azidoisophthalate, TzDB = 4,4′-(1,2,4,5-tetrazine-3,6-diyl)dibenzoate), where the porous structure can be transformed by chemical reactions between host framework and guests, therefore exhibit intriguing self-accelerating absorption and domino-type porous transformation, respectively[25,26]. These results encourage the further exploration of superior flexible MOFs whose porous structure can be reformed by unusual host-guest interactions. In general, MOFs composed of conformation-variable ligands are susceptible to the chemical environmental variations induced by guest adsorption[8,17,27–29], particularly when the local rotation or reorientation of ligands can be propagated in the global structure through cooperative motions of the framework. Such flexible features can be enhanced in two-dimensional (2D) framework materials as layer architectures that are assembled further via weak interlayer intermolecular interactions possess inherent structural ductility[30–33].

Structural transformations may affect the electronic configuration of magnetic centers, leading to switchable magnetic properties. The adsorption-responsive magnetic functions of SPCs broaden the prospect of applications and provide a new way to explore the mechanism of host-guest interactions through coupling effects[34–36]. A $CO_2$-induced paramagnetic–to–ferrimagnetic variation was recently demonstrated in a porous compound, [{Ru$_2$(F$_3$PhCO$_2$)$_4$}$_2$TCNQ(OEt)$_2$]·3DCM (TCNQ(OEt)$_2$ = 2,5-diethoxy-7,7,8,8-tetracyanoquinodimethane; F$_3$PhCO$_2^{2-}$ = 2,4,6-trifluorobenzoate; DCM = dichloromethane), as a result of electron transfer and a structural transition involved in the adsorption of $CO_2$ molecules[37]. In addition to radical molecules, the spin-crossover (SCO) compounds of $3d^4$–$3d^7$ metal ions, whose spin states can be interconverted between high-spin (HS) and low-spin (LS) states in response to external stimuli, are also susceptible to the structural changes in the coordination geometry of metal centers[38–41]. Therefore, the intercorrelation between the guests and SCO frameworks potentially leads to intriguing magnetic switching that directly entangles the structural transformations involved in guest adsorption.

In this work, a 2D SCO compound, [Fe$^{II}$(prentrz)$_2$Pd$^{II}$(CN)$_4$] (**1**), which possesses a Hofmann-type structure that usually has a porous structure[42–47], is designed and synthesized using a conformationally flexible organic molecule, prentrz, as an axial ligand. The difference from typical SPCs, whose pores expand with pressure increases (Fig. 1)[1,2], is the unusual pore rearrangement accompanied by simultaneous pore opening and closing caused by the non-uniform pedal motion of the prentrz ligand around its N−N single bond and global crumpling/unfolding of the layer structure in response to the adsorption of guest water molecules. The atypical pore transformation is accompanied by a distinct adsorption process in which to be activated first by long-time exposure in saturated steam to accomplish the slow dynamics between nqp and lcp structures. The direct variant multistep spin transition of the nqp and lcp phases is a result of the flexible framework and the host-guest interactions.

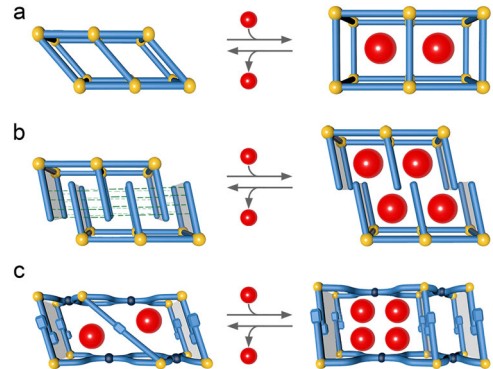

**Fig. 1 Guest adsorption-induced lattice transformations of SPCs.**
**a** Framework breathing induced by the hinging motion of organic ligands around the metal nodes. **b** Gate-opening/closing induced by sub-lattice displacive motion. **c** Framework breathing associated with a pore rearrangement induced by a non-uniform molecular reorientation.

## Results

**Crystal structures of the lcp and nqp phases**. The pristine single crystals were synthesized by the slow diffusion of a $CH_3OH$ solution of prentrz and $K_2$[Pd(CN)$_4$] into a $H_2O$ solution of FeSO$_4$·7$H_2O$. Crystals of the lcp phase **1·9/2$H_2O$** were obtained by stabilizing the as-grown crystals that contained $H_2O$ and MeOH mixture guests in air for 3 h (Supplementary Fig. 1). Single-crystal X-ray diffraction (SC-XRD) analysis showed that the compound crystallizes in the triclinic P$\bar{1}$ space group (Supplementary Table 1). In the crystal, Fe$^{II}$ ions are equatorially coordinated by four N atoms from four [Pd(CN)$_4$]$^{2-}$ ligands that bridge the metal centers in a 2D network with two axial positions occupied by two prentrz ligands (Fig. 2 and Supplementary Fig. 2). The asymmetric unit of the framework in lcp phase contains two crystallographically independent Fe$^{II}$ ions lying in the inversion centers, two prentrz ligands with different conformations (conformer 1 for Fe1 and conformer 2 for Fe2), and one [Pd(CN)$_4$]$^{2-}$ bridging ligand (Supplementary Fig. 3a). The 2D coordination networks were assembled further via intermolecular interactions into a 3D porous structure with channel pores (13.2 × 4.4 Å$^2$) extending along the crystallographic a-axis (Fig. 2c and Supplementary Fig. 4, regardless of the van der Waals radii)[48,49]. According to SC-XRD, elemental and thermogravimetric analyses, the pores accommodate 9/2 $H_2O$ molecules per Fe$^{II}$ atom (Supplementary Fig. 5a). The porous nature of framework was supported by a $CO_2$-adsorption measurement (Supplementary Fig. 6).

The crystals of the nqp phase **1·4/3$H_2O$** were obtained by heating **1·9/2$H_2O$** to 433 K for 48 h under vacuum and cooling in air to rehydration. Upon water desorption and re-adsorption, the structural transformation can be fully characterized at the atomic level by SC-XRD owing to the single-crystal-to-single-crystal nature. In this hydrous crystal, the basic structural features are retained, and the P$\bar{1}$ space group is the same as that of the **1·9/2$H_2O$**, while the cell parameters and direction undergo significant changes (Fig. 2d and Supplementary Table 2). Accordingly, one and a half Fe$^{II}$ ions (Fe1 lies in the inversion center), three prentrz ligands (conformer 1 for Fe1 and conformers 2 and 2′ for Fe2), and one and a half [Pd(CN)$_4$]$^{2-}$ ligands are found in the asymmetric unit of the framework (Supplementary Fig. 3b). The most significant structural changes occur in the prentrz ligands, which perform pedal rotation and reorientation upon water desorption (Fig. 2d)[50]. The directional shift of the axial ligands induces a large contraction in the pore size that converts the large channel pores in **1·9/2$H_2O$** to the narrow quasi-discrete pores in **1·4/3$H_2O$**, with a decrease in interlayer distance from 14.618(3) Å

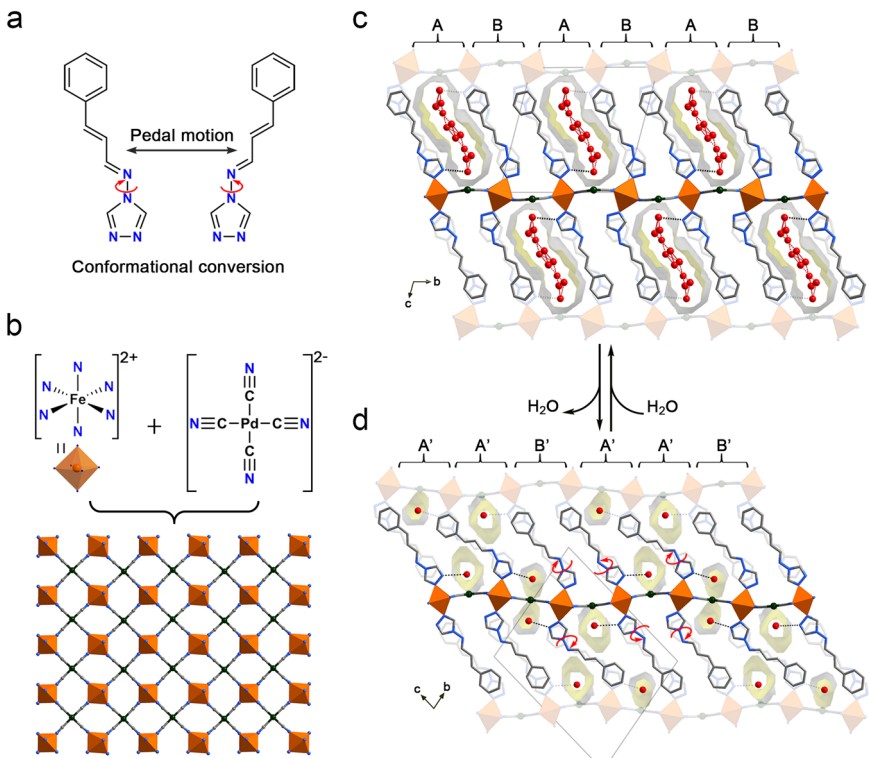

**Fig. 2 Water adsorption-induced reversible single-crystal to single-crystal transformation. a** The conformational flexible prentrz molecules are employed as axial ligands. **b** The 2D network constructed from Fe−N(≡ C − Pd) coordination bonds. The layer structures are further assembled into a 3D framework via interlaminar molecular interactions. **c** Single-crystal structure of the lcp phase **1**·9/2H₂O. **d** Single-crystal structure of the nqp phase **1**·4/3H₂O. The water adsorption-actuated reorientations and pedal rotations of the axial prentrz ligands repartition the lattice structure, manifesting as a narrow-pore to large-pore transformation accompanied by a rearrangement of the pore configuration from A'A'B'A'A'B' mode in nqp phase **1**·4/3H₂O to ABABAB mode in lcp phase **1**·9/2H₂O (A: open channel, B: closed channel). Therefore, the opening (A'/B' to A) and closing (A' to B) of pores coincide during water adsorption. According to SC-XRD analyses, 4/3 H₂O and 9/2 H₂O per Fe$^{II}$ ion accommodate in the pore of **1**·4/3H₂O and **1**·9/2H₂O, respectively. The dotted lines represent the hydrogen-bond interactions. Fe, orange; C, gray; N, blue; O, red. The hydrogen atoms were omitted for clarity.

to 13.296(5) Å (Supplementary Fig. 7). The narrow pores of this hydrous crystal accommodate 4/3 H₂O molecules per Fe$^{II}$ atom according to the SC-XRD, elemental and thermogravimetric analyses (referred to **1**·4/3H₂O, Supplementary Fig. 5b). Although Powder X-ray diffraction (PXRD) experiments revealed that in situ generated **1** by heating **1**·9/2H₂O at 433 K for 48 h under vacuum possessed the similar contracted structure as that of nqp phase **1**·4/3H₂O, **1** and **1**·4/3H₂O were not exactly the same phases (Supplementary Fig. 8). However, an attempt to obtain the crystal structure of completely dehydrated crystal of **1** failed because of the poor diffraction. The phase purity of lcp phase **1**·9/2H₂O and nqp phase **1**·4/3H₂O was confirmed by Rietveld refinement pattern (Supplementary Fig. 9).

During the dehydration process, the axial prentrz ligands undergo different molecular motions and the inconsonant molecular reorientations lead to a rearrangement in the pore configuration, i.e., the channel state of the ABABAB mode in the lcp phase is converted to the A'A'B'A'A'B' mode in the nqp phase (A: open channel; B: close channel). Although numerous flexible MOFs manifesting structural transformations in response to guest adsorption/desorption have been reported[2,26], such a guest adsorption-actuated statistical rearrangement of the pore configuration accompanied by simultaneous opening and closing of pores is very uncommon in porous materials.

**Water adsorption isotherms**. The adsorption properties of this material were examined by performing vapor adsorption/desorption isotherm measurements under different activating conditions.

The **1**·9/2H₂O was activated in situ under vacuum at 433 K for 48 h to ensure that the water molecules were completely removed. After that, the water adsorption isotherm started with a flat plateau in $P/P_0 = 0$–$0.08$ range. The adsorption behavior may indicate structural transformation from closed framework to nqp framework, corresponding to the transition from **1** to **1**·4/3H₂O (Fig. 3a, up). The uptake saturated at $P/P_0 = 0.89$, corresponding to the adsorption of ~1.4 H₂O per Fe$^{II}$. This result is consistent with the 4/3 H₂O molecules identified by SC-XRD of **1**·4/3H₂O, suggesting no nqp–to–lcp gate-opening transition occurred in this sample. However, water adsorption was enhanced significantly when the above anhydrous **1** or **1**·4/3H₂O was left in the steam atmosphere ($P/P_0 = 0.9$) for 4 days and was then activated under vacuum at 303 K for 6 h (or directly activated **1**·9/2H₂O at 298 K for 6 h). As shown in Fig. 3a (middle), the uptake instantly reached ~1.1 H₂O per Fe$^{II}$ at $P/P_0 = 0.02$ and then increased slightly to 1.78 H₂O per Fe$^{II}$ at $P/P_0 = 0.64$, after which the adsorption suddenly reached 4.5 H₂O per Fe$^{II}$ at $P/P_0 = 0.86$. The absorption of 4.5 H₂O per Fe$^{II}$ at $P/P_0 = 0.86$ is consistent with the single-crystal structural analyses of **1**·9/2H₂O, indicating the gate-opening transition of this sample.

The activation-method dependent water adsorption was verified by a continuous adsorption-desorption-cycles measurement. As shown in Fig. 3b, when the sample prepared from a long-time heating (14 h) at 433 K under vacuum condition was placed in a steam atmosphere with $P/P_0 = 0.9$, it underwent two-step water adsorption: a rapid partially water adsorption of ~1.7 H₂O per Fe$^{II}$, and later a very slow water adsorption (97 h) of ~4.5 H₂O per Fe$^{II}$ that reflected the **1**·4/3H₂O-to-**1**·9/2H₂O gate-

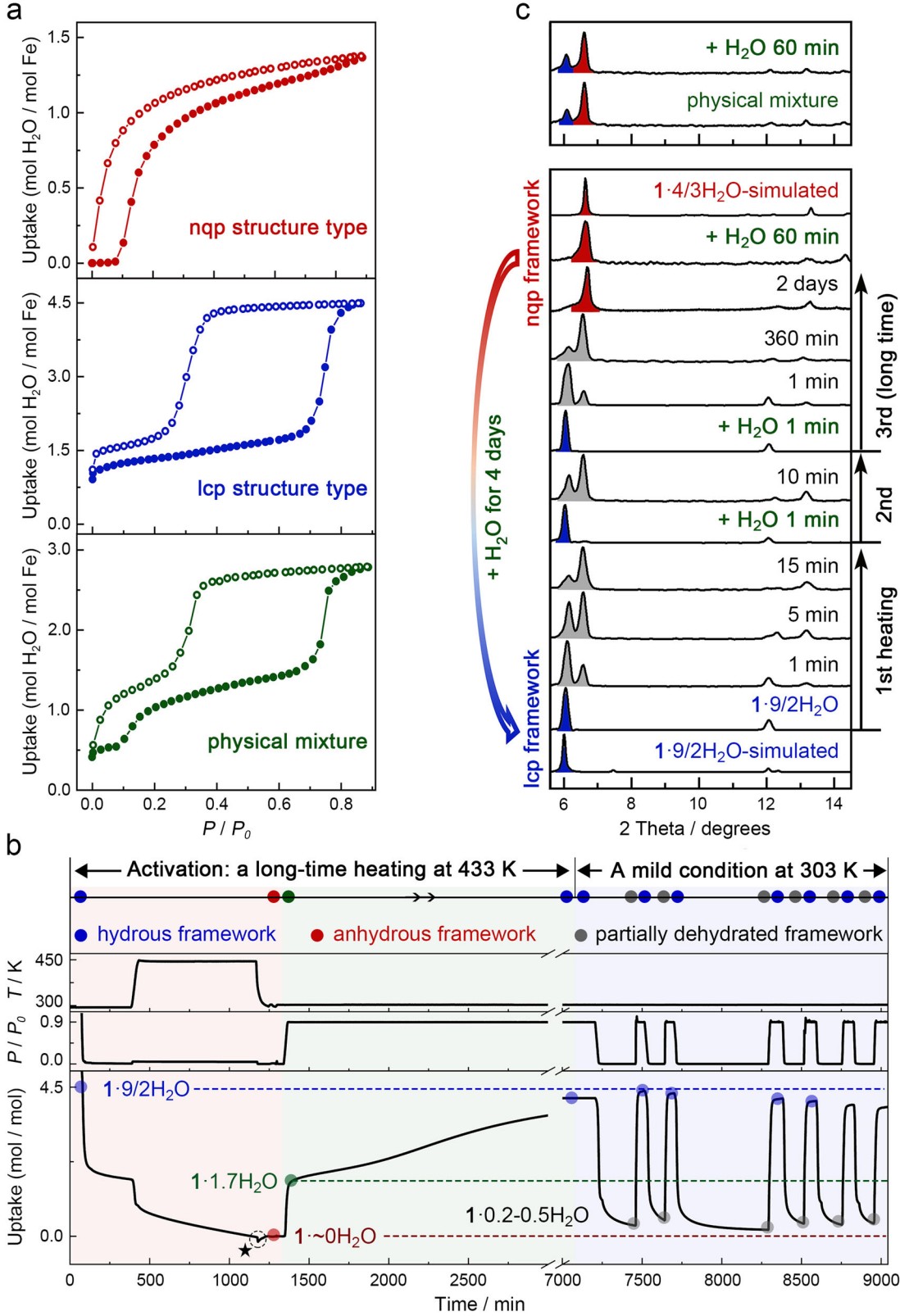

opening structural change. When the regained sample of **1**·9/ 2H$_2$O was not dehydrated thoroughly under a mild condition (303 K under vacuum for 10 h) with ~0.2–0.5 H$_2$O per Fe$^{II}$ retained, the sample manifested an abrupt one-step water adsorption (10 min) to ~4.5 H$_2$O per Fe$^{II}$ in a steam atmosphere with $P/P_0 = 0.9$. The distinct adsorption properties suggest that the samples prepared from different activating methods undergo

different adsorption processes and imply that a small amount of lattice water molecules in structure have a very large impact on the flexibility of the whole framework[51]. The two-step water adsorption was confirmed by a another sample activated at 400 K under vacuum for 22 h (Supplementary Fig. 10). Notably, the PXRD pattern after two isothermal vapor ad/de-sorption or the adsorption-desorption-cycles measurement revealed that the

**Fig. 3 Water adsorption isotherms and corresponding structural transformations probed by PXRD. a** Isothermal vapor adsorption (solid red) and desorption (open red) of the sample activated at 433 K under vacuum for 48 h (up), isothermal vapor adsorption (solid blue) and desorption (open blue) of the sample activated at 298 K under vacuum (middle) and isothermal vapor adsorption (solid green) and desorption (open green) of the physical mixture (nqp and lcp phases) at 298 K under vacuum (down). **b** The continuous water adsorption-desorption cycles in different activated conditions. The abnormal drop of "Uptake" line marked by star at 1200 min is ascribed to the change of heating source (Furnace and water bath). **c** PXRD patterns upon water adsorption and desorption. $1 \cdot 9/2H_2O$ loses water rapidly upon heating (323 K), which leads to a partially dehydrated sample that can return to the lcp phase immediately by spraying with a water mist. Such rapid water adsorption of the partially dehydrated sample was performed two times to confirm the phenomenon. However, the completely dehydrated sample **1** prepared from long-time heating (433 K) under vacuum can rehydrate to nqp phase $1 \cdot 4/3H_2O$ within 60 min, while cannot recover to lcp phase smoothly upon water adsorption. In the measured sample, it took 4 days after leaving the **1** or $1 \cdot 4/3H_2O$ in saturated steam. The PXRD patterns of the physical mixture of two phases did not show any significant change after spraying the water mist for 60 min. The gray, blue and red peaks denote the character peaks of structures of partially dehydrated, lcp and nqp framework, respectively.

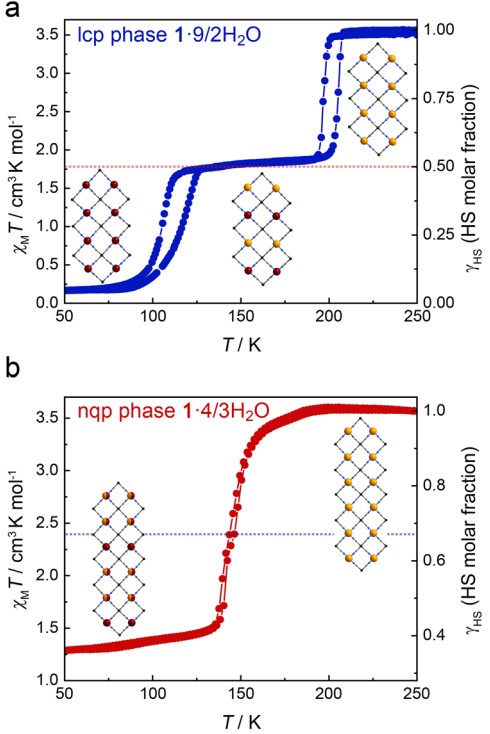

**Fig. 4 Temperature dependence of the $\chi_M T$ curves of lcp phase $1 \cdot 9/2H_2O$ and nqp phase $1 \cdot 4/3H_2O$ upon heating and cooling.** The two-step SCO with a large plateau of lcp phase $1 \cdot 9/2H_2O$ (**a**) and a two-step SCO without plateau of nqp phase $1 \cdot 4/3H_2O$ (**b**).

crystallinity and framework of nqp and lcp structure type samples can both be revived by rehydration, and further reflected the different structural flexibilities between anhydrous **1**, nqp phase $1 \cdot 4/3H_2O$ and lcp phase $1 \cdot 9/2H_2O$ (Supplementary Fig. 11).

**Structural transformation investigated by powder X-ray diffraction (PXRD) and micro-Raman spectroscopy.** To understand the different water adsorption properties, in situ PXRD was performed to probe the structural differences in the samples prepared from different activating methods. When the crystal sample of $1 \cdot 9/2H_2O$ in lcp phase was subjected to mild activation, i.e., thermal treatment (323 K) or exposed to vacuum, new peaks appeared in the higher-angle region within 1 min and increased gradually with time, indicating the contraction of partially dehydrated framework (Fig. 3c and Supplementary Figs. 5, 12–14). The lcp phase could be recovered immediately by spraying water mist on the partially dehydrated samples. The desorption-adsorption process had been repeated two times to confirm the rapid water adsorption of the sample (Fig. 3c and Supplementary Fig. 15). The complete lcp–to–nqp structural

transition of all samples was unexpectedly difficult, since the characteristic peaks of partially dehydrated framework remained even after the sample was heated at 433 K under vacuum for 18 h (Supplementary Fig. 16). Therefore, the sample was heated to 433 K under vacuum for 48 h in the third cycle to obtain completely dehydrated sample of **1**. However, the resulting anhydrous **1** could rehydrate to nqp phase $1 \cdot 4/3H_2O$ within 60 min, while cannot recover to lcp phase smoothly upon water adsorption after being sprayed with water mist as in the former two cycles. Similar to the water-adsorption measurements, the backward nqp–to–lcp structural transition could only be achieved by leaving the nqp sample in saturated steam for a long-time (~four days in the measured sample, see Fig. 3c and Supplementary Figs. 17 and 18). These results suggest that the lcp structural characteristics of partially dehydrated framework can persist in the sample prepared under mild activating conditions, and such flexible framework plays a vital role in the doubled water adsorption[24,51–54]. Further experiments confirmed that the partially dehydrated samples rather than a physical mixture of two pure phases because the PXRD patterns of a physical mixture of the lcp phase $1 \cdot 9/2H_2O$ and nqp phase $1 \cdot 4/3H_2O$ did not show any significant change after spraying the water mist (Fig. 3c).

In order to further verify that the activation-method dependent adsorption properties are related to structural difference between nqp and partially dehydrated frameworks, micro-Raman spectra experiments (spot diameter ~2 μm) were performed on single crystals[55,56]. As shown in Supplementary Fig. 19, the Raman spectrums of lcp and nqp phases present delicate difference in two regions: the bands around 1170 cm$^{-1}$ (C–C strech modes of prentrz ligands) and 1140 cm$^{-1}$ (CH-bending of prentrz ligands)[57]. In the nqp single crystal, the bands at around 1170 and 1140 cm$^{-1}$ persisted up to 3 days in saturated steam, and then coverted to the characteristic bands of lcp structure type. For comparison, the bands at around 1140 cm$^{-1}$ remained unchanged as the lcp single crystal $1 \cdot 9/2H_2O$ under mild activation (a constant $N_2$ gas), suggesting the lcp structural characteristics of partially dehydrated framework. The Raman spectrum of partially dehydrated single crystal can change to that of lcp struture type within 10 min in a humidity of 80%. The distinct responses of nqp and partially dehydrated single crystals in the micro-Raman spectra verify the structural dependent water adsorption property in this material.

**SCO properties of the samples in different phases.** SCO properties of the magnetic centers in the 2D Hofmann structure are usually sensitive to the geometric variations of the coordination sphere and guest variation[58]. Therefore, we measured the temperature-dependent magnetic susceptibilities of $1 \cdot 9/2H_2O$, $1 \cdot 4/3H_2O$, and partially dehydrated samples with different heating time to investigate magnetic switchings corresponding to water adsorption and desorption (Fig. 4 and Supplementary Fig. 22). Figure 4a shows the $\chi_M T$ versus $T$ plot of $1 \cdot 9/2H_2O$, in

which the $\chi_M T$ values are 3.48–3.57 cm$^3$ K mol$^{-1}$ at $210 - 250$ K and $0.16 - 0.17$ cm$^3$ K mol$^{-1}$ at $50 - 65$ K, corresponding to the full HS state ($\gamma_{HS} = 1$) and full LS state ($\gamma_{HS} = 0$), respectively. During the SCO process, a stepwise spin transition with a wide intermediate plateau (74 K) is observed over a temperature range of $118 - 192$ K. The $\chi_M T$ values of $1.72 - 1.89$ cm$^3$ K mol$^{-1}$ at this intermediate temperature range indicate that the sample is in the HS$_{0.5}$LS$_{0.5}$ state. The two-step spin transition is accompanied by substantial thermal hysteresis loops of *ca.* 9 K ($T_{c1}\!\downarrow = 196$ K, $T_{c1}\!\uparrow = 205$ K) and 15 K ($T_{c2}\!\downarrow = 166$ K, $T_{c2}\!\uparrow = 181$ K). Such a hysteretic spin transition is usually caused by strong cooperative interactions among the magnetic centers[41,42]. The shifts in the Fe$-$N bond lengths confirm the spin transition of Fe$^{II}$ ions in **1**·9/2H$_2$O. The average Fe1$-$N bond lengths are 2.167(5) Å at 250 K and 1.964(5) Å at 150 K, respectively, indicating that a complete HS-LS transition of Fe1 occurs in the first step spin transition upon cooling (Supplementary Figs. 23–25 and Table 3)[40]. The average Fe2$-$N bond length shortens from 2.160(4) Å at 150 K to 1.970(4) Å at 85 K, showing that Fe2 undergoes a complete spin transition at the lower temperature range. Therefore, the two-step spin transition of **1**·9/2H$_2$O is accounted for by the two metal centers in the asymmetric unit, respectively.

The $\chi_M T$ values of **1**·4/3H$_2$O are the same as those of **1**·9/2H$_2$O over the temperature range of $200 - 250$ K, suggesting a full HS state ($\gamma_{HS} = 1$) of **1**·4/3H$_2$O at high-temperature range (Fig. 4b). Upon cooling, the $\chi_M T$ value shows a slight decrease until 160 K. It then decreases abruptly and finally reaches 1.28 cm$^3$ K mol$^{-1}$ at 50 K. An obscure inflection point at 147 K indicates that the **1**·4/3H$_2$O sample undergoes a two-step but incomplete spin transition (Supplementary Fig. 26). The average Fe1 $-$ N bond lengths of 2.134(9) Å at 250 K and 1.949(6) Å at 100 K (Supplementary Table 4), respectively, suggest that Fe1 ion in **1**·4/3H$_2$O undergoes an complete SCO. While for Fe2 ion, an incomplete SCO is concluded as the average Fe2 $-$ N bond lengths shorten from 2.144(10) Å at 250 K to 2.062(7) Å at 100 K. The $\chi_M T$ value shows unchange at 50 K with different scan rates (10, 5, 2, and 1 K min$^{-1}$), suggesting the incomplete spin transition of Fe2 was not kinetically trapped (Supplementary Fig. 27). The water adsorption-induced SCO variations can be attributed to the geometric changes in the coordination spheres of Fe$^{II}$ ions and the shifts in the intermolecular interactions, which in turn affect the electron states of the coordinated ligands[38,40–42,58]. For the SCO properties of partially dehydrated samples, a hysteretic spin transition at $133 - 147$ K emerged when **1**·9/2H$_2$O was partially desolvated. Upon further water molecules losing, a series of variations in the total SCO properties were accompanied by the intermediate hysteresis loop increasing, resulting in the three- and four-step SCO. Though the SCO variation of partially dehydrated samples cannot be ruled out as the linear combination of dehydrated and non-dehydrated particles, the partially dehydrated samples with relatively minimal water molecules showed the incomplete SCO with the similar $T_{1/2}$ as that of nqp phase **1**·4/3H$_2$O, suggesting that the host-guest interaction affecting SCO behavior is determined by the amount of water molecules and dynamic framework[59–61].

## Discussion

The slow lcp-nqp dynamics is compared to illuminate the microscopic structural origin of the atypical pore rearrangement in response to water adsorption-desorption process. As shown in the superimposed drawing of the frameworks in the lcp (**1**·9/2H$_2$O) and nqp (**1**·4/3H$_2$O) phases, the axial prentrz ligands that underpin the 3D framework undergo significant variations (Fig. 5). In particular, the ligands III, IV, and V in the nqp phase perform in-plane pedal motions around their N $-$ N single

bonds. Moreover, the Fe$-$N coordination bonds of prentrz in II, III, IV, and V undergo distinct shifts in their bond direction during water adsorption (Supplementary Fig. 28). As a comprehensive result, ligands II and IV exhibit the most remarkable reorientations in their principal molecular directions, while those of ligands I, III, V, and VI show minor changes. The non-uniform directional shifts of the axial linear ligands repartition the lattice structure (Fig. 5b). Remarkably, the large reorientation of ligand II during water adsorption enlarges channel A$_1$' and closes channel A$_2$'. The reorientation of ligand IV opens channel B$_1$' and closes the narrow channel A$_3$' in the nqp phase. Therefore, the A'A'B'A'A'B' array of the pore configuration in the nqp phase transforms to the ABABAB mode in the lcp phases (Fig. 5c and Supplementary Fig. 29). During water adsorption, the 2D networks connected by Fe$-$N($\equiv$C$-$Pd) coordination bonds also demonstrate remarkable structural flexibility that manifest as a crumpling/unfolding of the layer structure (Fig. 5d and Supplementary Fig. 30). The fluctuations of the layer structure assist in the directional shifts of the Fe$-$N coordination bonds of the axial ligands. The 3D framework of this compound is secured by the $\pi\cdots\pi$ interactions between the prentrz ligands in the adjacent layers (Supplementary Fig. 31). The substantial interlayer interactions prevent the large relative displacement of the layer structures. Consequently, the adsorption-induced structural transformation manifests a reorientation of axial ligands accompanied by a minor sliding motion of the coordination layers (Supplementary Fig. 32). The lattice structure repartitioned by the inconsonant reorientation of axial ligands in this soft porous compound is different from the typical adsorption-induced structural transformations, such as the lattice displacement or hinging motion of organic ligands around the metal nodes (Fig. 1), therefore, leading to an unusual pore rearrangement in this compound, rather than only the framework breathing in typical flexible MOFs[1,2,7,21,37].

During the pore rearrangement from the nqp phase to the lcp phase upon adsorption, the water molecules residing in channels A$_2$' and A$_3$' of **1**·4/3H$_2$O need to be evacuated to close the pores (Fig. 5c and Supplementary Movie 1). Such simultaneous pore opening and closing increases the energy barrier from the nqp phase to the lcp phase. This is particular in the present case because the excluded water molecules exhibit strong hydrogen bonds with the N atoms of the framework with O$\cdots$N distances of 2.754(2) Å and 2.908(3) Å (250 K, Fig. 5c and Supplementary Fig. 31 and Table 5). Hence, the onset of nqp–to–lcp structural transformation is extremely time consuming[2]. However, once the lcp domains are generated, the high energy barrier should be reduced significantly by the interfacial effect between the expanded lcp and contracted nqp phases. The interfacial interactions are effectively transmitted by the fluctuations of the Fe$-$N($\equiv$C$-$Pd) coordination bonds, slippage of the layer structures, and $\pi\cdots\pi$ interactions between the prentrz ligands in different layers (Supplementary Fig. 33 and Supplementary Movie 1). Therefore, smooth gate opening-related water adsorption can be realized in self-accelerating structural transformation[25].

Density functional theory calculations were performed to investigate the energy variations upon water adsorption and support the experimental results (Fig. 6 and Supplementary Fig. 34). The adsorption structures were optimized on the unit cells of the nqp and lcp phases. Owing to the different asymmetric units of the nqp and lcp phases, the number of atoms (2nqp = 3lcp) was balanced to make direct energetic comparisons (Supplementary Fig. 35). For a completely desolvated sample, the nqp phase is energetically preferred compared to the lcp phase, and it can be stabilized further by 173 kcal mol$^{-1}$ when four water molecules are loaded in the unit cell. This calculation is

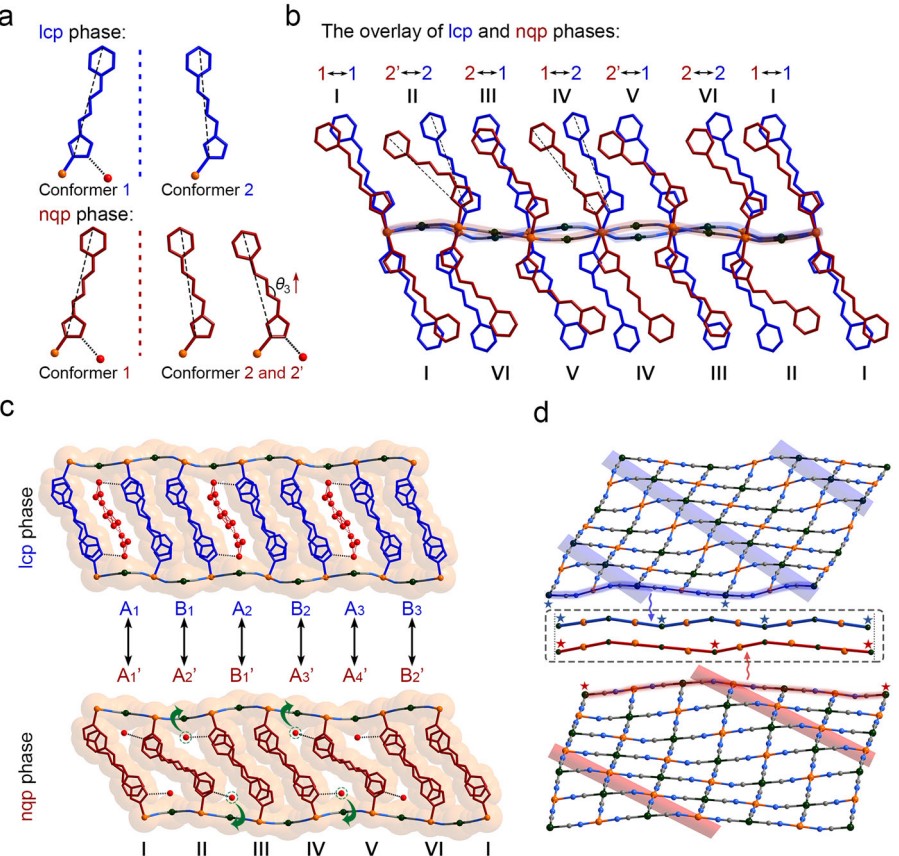

**Fig. 5 Detailed structural transformation upon water adsorption. a** Conformations of flexible prentrz ligands in the lcp (**1**·9/2H$_2$O) and nqp (**1**·4/3H$_2$O) phases. **b** Local pedal rotations of prentrz molecules and the directional shifts of axial Fe−N coordination bonds. The ligands III, IV, and V perform in-plane pedal rotations around their N−N single bonds, and ligands II, III, IV, and V undergo distinct directional shifts with angles of 12.9°, 22.1°, 37.8°, and 18.0°, respectively. The pedal rotations and the reorientations of Fe−N bonds have opposite influences on the principal directions of axial ligands. Consequently, ligands II and IV show the most significant shifts in their molecular orientations (as dashed lines indicated). **c** The directional shift of ligand II upon adsorption expands channel A$_1$' and closes channel A$_2$', while the directional shift of ligand IV opens channel B$_1$' and closes channel A$_3$'. Accordingly, the four water molecules accommodated in pore A$_2$' and A$_3$' of nqp phase must be evacuated to facilitate the nqp–to–lcp lattice transformation. **d** The coordination layer demonstrated a remarkable crumpling/unfolding motion to assist the reorientation of the Fe−N coordination bonds. The structures of lcp (**1**·9/2H$_2$O) and nqp (**1**·4/3H$_2$O) phases are drawn in blue and red, respectively.

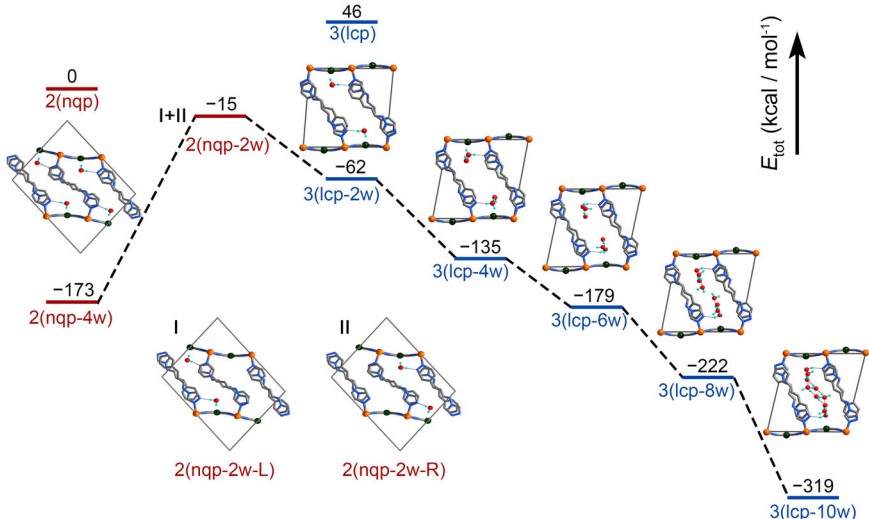

**Fig. 6 Energy diagram of water adsorption.** The adsorption-induced nqp–to–lcp lattice transformation experiences simultaneous pore opening and closing (nqp-4w to nqp-2w state) with a large energy requirement of 158 kcal mol$^{-1}$, after which nqp-2w-L/R can be converted to lcp-2w to reduce the lattice energy.

consistent with the experimental observation that 4/3 $H_2O$ per $Fe^{II}$ ion can be adsorbed instantly at low pressure. In the nqp–to–lcp process, upon further water adsorption, the sample experiences simultaneous pore opening and closing, where two water molecules in the channel need to be evacuated to accomplish the pore rearrangement. The nqp-4w-to–nqp-2w process is energy consuming with an entropy gain. The uphill energy of 158 kcal mol$^{-1}$ is remarkably larger than the energy variations involved in the guest-induced structural transformations of typical FMOFs[9,15,18–20,62]. Once nqp-2w-L/R form, the structures are converted to lcp-2w because the latter is 47 kcal mol$^{-1}$ more stable than nqp-2w-L/R. The lcp-2w makes room for continuously adsorbing water molecules (Supplementary Fig. 34). Therefore, energy-favored adsorption proceeds with a decrease in energy for lcp-2w → lcp-4w → lcp-6w → lcp-8w → lcp-10w (ca. 9/2 $H_2O$ per $Fe^{II}$ ion). The high energy barrier in the nqp-4w-to–nqp-2w process should play important role in the activation-method dependent water adsorption of this compound.

In conclusion, a 2D soft magnetic porous single-crystal compound was prepared utilizing the conformational flexibility of the prentrz ligand and the fluctuation of the 2D coordination network. The water adsorption-induced, non-uniform pedal rotations of the axial ligands accompanied with crumpling/unfolding of the 2D layer structure repartition the lattice structure, which manifests as framework breathing with a rearrangement of the pore configuration. The simultaneous pore opening and closing involved in the pore rearrangement leads to slow nqp–lcp dynamics, where nqp phase need to long-time exposure in saturated steam to accomplish the nqp–to–lcp gate-opening adsorption. Moreover, the structural transformation of the magnetic framework shifts the SCO properties of the $Fe^{II}$ magnetic centers. This study presents a adsorption-related pore transformation accompanied by activation-method dependent adsorption. Such an exotic property may be used in materials for gas adsorption, actuation, and sensing.

## Methods

**[$Fe^{II}$(prentrz)$_2$Pd$^{II}$(CN)$_4$]·9/2$H_2O$ (1·9/2$H_2O$).** An aqueous solution (3 mL) containing FeSO$_4$·7$H_2O$ (0.05 mmol, 13.90 mg) and ascorbic acid (2.0 mg) was placed at the bottom of a test tube, a buffer solution of $H_2O$ and $CH_3OH$ (1:1, v/v, 2 mL) was carefully layered on the top of the solution. After then, a $CH_3OH$ solution (3 mL) that contained prentrz ligand (0.10 mmol, 19.82 mg) and K$_2$[Pd(CN)$_4$] (0.03 mmol, 8.66 mg) was added as the third layer. After 2 weeks, yellow plate-like crystals were grown. The crystals of 1·9/2$H_2O$ were collected after stabilized in the air condition for 3 h (yield: 15.42 mg, ~ 68.9% based on K$_2$[Pd(CN)$_4$]). The amount of solvent molecules in the crystal was estimated by single-crystal X-ray diffraction, elemental and thermogravimetric analyses. FTIR (KBr pellets, cm$^{-1}$): 415, 491, 506, 619, 691, 752, 984, 1058, 1178, 1450, 1515, 1629, 2164, 3130, 3387. Elemental analysis calcd (%) for C$_{52}$H$_{58}$O$_9$N$_{24}$Fe$_2$Pd$_2$: C 41.98, H 3.93, N 22.60; found: C 42.11, H 3.82, N 22.51.

**[$Fe^{II}$(prentrz)$_2$Pd$^{II}$(CN)$_4$]·4/3$H_2O$ (1·4/3$H_2O$).** The crystals of 1·4/3$H_2O$ were prepared by heating the sample of 1·9/2$H_2O$ at 433 K under vacuum for 48 h and then cooling in the air condition. The amount of solvent molecules in the crystal was estimated by single-crystal X-ray diffraction, elemental and thermogravimetric analyses. Elemental analysis calcd (%) for C$_{78}$H$_{68}$O$_4$N$_{36}$Fe$_3$Pd$_3$: C 45.47, H 3.33, N 24.47; found: C 45.39, H 3.30, N 24.41.

**Sorption experiments.** The vapor adsorption isotherms were recorded on a Hiden-Isochema Intelligent Gravimetric Analyzer (IGA 100 C) at 298 K. In the first measurement, the 1·9/2$H_2O$ sample was activated in situ at 433 K under vacuum for 48 h and then measured at 298 K. The sample chamber was then pressurized to the set pressure and allowed to equilibrate for 120 min before moving to the next pressure point. After the first measurement, the sample was left in the chamber at the steam environment ($P/P_0 = 0.9$) for 4 days to regain the 1·9/2$H_2O$. In the second measurement, the 1·9/2$H_2O$ sample was activated at 298 K under vacuum for 6 h. The adsorption isotherm measurements were then performed under the same test conditions. The continuous adsorption-desorption-cycles measurement began with the the sample was heated from room temperature to 433 K and maintained for 14 h. After the activated process, the adsorption process was

performed at steam environment (38.2 mbar, $P/P_0 = 0.9$) for 4 days to recover the 1·9/2$H_2O$. The 2nd − 7th adsorption-desorption measurement were followed under mild activated conditions: The desorption processes were performed at 303 K under vacuum for 4, 2, 10, 2, 3 and 2 h, respectively. And, the adsorption processes reached saturation by about 10−20 min at steam environment (38.2 mbar, $P/P_0 = 0.9$).

**Powder X-ray diffraction (PXRD).** The variable- and room-temperature PXRD data were recorded on a PANalytical diffractometer with Cu Kα radiation equipped with different temperature control parts: (1) The Anton Paar HTK 1200 accessory was designed for PXRD in reflection and transmission geometries with environmental heating of the sample up to 1473 K in air. The thermocouple was placed right underneath the round sample table (diameter ~1.5 cm, thickness ~0.5 mm and aluminium oxide material). (2) The Anton Paar TTK 450 chamber enabled in situ PXRD measurements in both reflection and transmission with the environmental temperature range from 100 K to 450 K under vacuum (liquid nitrogen cooling). The thermocouple was placed underneath the 1.4 × 1.8 cm$^2$ sample table (thickness ~1 mm and stainless steel material). For PXRD measurements that did not require temperature control part, the sample was loaded on the glass sample table with circular groove (diameter ~2 cm, thickness ~1 mm).

Temperature-dependent synchrotron PXRD data were collected at the BL02B2 beamline (Spring-8, Japan, λ = 0.4500 Å). The crystalline powder of nqp phase 1·4/3$H_2O$ in a silica glass capillary (0.5 mm diameter) was measured at 250 K. The crystalline powder of lcp phase 1·9/2$H_2O$ in a silica glass capillary (0.5 mm diameter) was cooled and heated in the temperature range from 250 to 100 K (250−175−125−100−125−175−250 K). The obtained diffraction data were analysed by Rietveld refinement. The scale factor, background, peak shape, zero point, and lattice parameters were refined. The Pseudo-Voigt function was chosen to generate the peak shapes.

**Single-crystal X-ray diffraction (SC-XRD) analyses.** SC-XRD analyses of 1·9/2$H_2O$ and 1·4/3$H_2O$ were performed on a Rigaku Oxford XtaLAB PRO diffractometer equipped with graphite-monochromated Mo Kα radiation (λ = 0.71073 Å). The data of 1·9/2$H_2O$ were collected with one sample at 250, 150 and 85 K, successively. The data of 1·4/3$H_2O$ was collected at 250 K.

The crystal data of 1·9/2$H_2O$ and 1·4/3$H_2O$ was further collected with a Bruker/ARINAX MD2 diffractometer equipped with a MarCCD-300 detector (λ = 0.77484 Å) at the BL17B beam line station of the Shanghai Synchrotron Radiation Facility (SSRF). The data of 1·9/2$H_2O$ was collected at room temperature (293 K) and the data of 1·4/3$H_2O$ was collected at 100 K. 360 frames of each data were collected using ω-scans with oscillation angle of 1º, scan range of 0 to 360º, exposure time of 0.50 s per frame, and detector distance of 90 mm. Data were collected using the BlueIce software. The dataset was collected on the beamline equipped with 1-axis goniometer, which allows phi-scans. The orientation of the small-molecule crystal did not allow to collect 100 % of reflections in this regime, especially the low symmetry of crystal. Unit cell refinement and data reduction were performed using the HKL3000 software[63].

All the structures were solved by direct methods and further refined by full-matrix least-squares techniques on $F^2$ with SHELX program[64]. Non-hydrogen atoms were refined anisotropically, and the hydrogen atoms were generated geometrically and refined isotropically.

**Micro-Raman spectroscopy.** The micro-Raman spectra experiments were performed on a Renishaw InVia confocal Raman spectrometer equipped with a humidity control part and a Leica DMLM microscope. The more detailed information of the setup was described in previous literatures[65–67]. The excitation source was provided by an Argon ion laser with wavelength of 514.5 nm and output power of 5 mW (model Stellar-REN, Modu-Laser). A Raman notch filter (wavelength 514.5 nm) was used to remove the strong Rayleigh scattering. The spectral resolution of ~1 cm$^{-1}$ could be implemented by using a diffraction grating of 1800 g mm$^{-1}$ (grooves per millimeter). The single crystals of lcp phase 1·9/2$H_2O$ and nqp phase 1·4/3$H_2O$ were located on the cover glass by tape and chosen by the optical microscope with a 50 × objective lens (0.75 numerical aperture). The laser beam was focused on the selected region of single crystal with a spatial resolution of 6 μm. The spectra in the range 500−3000 cm$^{-1}$ were collected with three spectral scans at 10 s accumulation time. Fixed single crystals were dispensed onto PTFE substrate in the sample cell at a adjustable ambient humidity. Single crystal in partially dehydrated type was prepared by in situ drying the 1·9/2$H_2O$ under a nitrogen flow for 30 min.

**Magnetic measurement.** Magnetic measurements of lcp phase 1·9/2$H_2O$, nqp phase 1·4/3$H_2O$ and partially dehydrated samples were performed on a Quantum Design MPMS XL-7 magnetometer working in the 2−400 K temperature range with 2 K min$^{-1}$ sweeping rate under a magnetic field of 5000 Oe. The samples used for the magnetic measurements of Fig. 4 were prepared from the standard method described above, while those for the magnetic measurement of Supplementary Fig. 22 were prepared by heating the sample of 1·9/2$H_2O$ at 323 K for different times (~2 min for 4b, ~4 min for 4c, ~6 min for 4d, ~8 min for 4e). The prepared samples were tightly wrapped with a plastic film (2 × 2 cm$^2$) and fixed in a straw. After that, the sample was

loaded in the SQUID chamber at 250 K. The PXRD measurements were performed on the same samples to examine the corresponding states.

## Data availability

All data in this study are presented in the article and its Supplementary Information, which are also available from the authors upon request. The X-ray crystallographic coordinates for structures have been deposited at the Cambridge Crystallographic Data Center (CCDC) under deposition numbers CCDC 2093900 (1·4/3H₂O at 100 K), 2111045 (1·4/3H₂O at 250 K), 2124267 (1·9/2H₂O at 293 K), 2093901 (1·9/2H₂O at 250 K), 2093902 (1·9/2H₂O at 150 K), and 2093903 (1·9/2H₂O at 85 K). These data can be obtained free of charge via http://www.ccdc.cam.ac.uk/data_request/cif.

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

## Acknowledgements

This work was supported by the National Natural Science Foundation of China (Grants 21971016 and 92061106 (J.T.), 22071009 (Z.-S.Y.)) and Beijing Natural Science Foundation of China (Grant 2222028 (J.X.)). We thank Li-Xia Bao at the Analysis & Testing Center of Beijing Institute of Technology for the technical support of sorption measurements. The SC-XRD data was collected at BL17B beamline of the National Center for Protein Sciences at Shanghai Synchrotron Radiation Facility. The synchrotron PXRD experiments were performed at the BL44B2 of Spring-8 with the approval of the Japan Synchrotron Radiation Research Institute (JASRI; Proposal Nos. 2019b1415).

## Author contributions

J.-P.X., Z.-S.Y., and J.T. designed the study and wrote the manuscript. J.-P.X. synthesized the materials and performed the experimental measurements. Y.H., B.Z., and J.X. performed the DFT calculations. Z.-K.L. assisted in the magnetic measurements. Z.-S.Y. and J.T. supervised the research. All authors discussed the results and commented on the manuscript.

## Competing interests

The authors declare no competing interests.
