## [Peer Review File · Nature Communications]

A spin-crossover framework endowed with pore-adjustable behavior by slow structural dynamicsReviewer #1 (Remarks to the Author):

The authors describe a gate-opening adsorption in the Hofmann-type framework, which demonstrates water-induced variations of spin crossover. This is interesting on its own, but was previously presented for a Hofmann-type framework by Real et al. (Chem. Sci. 2020, 11, 11224-11234; reference 40). Moreover, authors postulate presence of the “phase-coexistence state of individual crystals”. Such an observation would be very valuable, but in my opinion authors do not present sufficient scientific evidence to support this claim. The design of the appropriate experiment is challenging, but it seems clear that it would require use of space-resolved techniques, while the authors performed most of the characterization on bulk powder samples (in addition to the reference 47, I encourage the authors to read the article by Neimark et al. in J. Phys. Chem. Lett. 2011, 2, 2033-2037). Moreover, manuscript contains numerous technical drawbacks that must be resolved before publication:

- 1) Single-crystal X-ray diffraction data for 1-lcp phase lead to structure solutions that show very large residual electron density in the proximity of the palladium ion. Authors use hundreds of OMIT instructions (more than 700 in case of 1-lcp at 150K) in order to artificially remove this electron density, which is a case of data manipulation that should never be performed. Instead, an adequate twin law should be applied to systematically remove electron density attributed to the twinned crystal or the experiment itself should be repeated (comments 3 and 5).
- 2) Crystal structure 1-nqp at 250 K is characterized by $R1 > 0.15$ and data completeness below 73%. In the CIF file authors claim: “The final refinement results with low $R1: 0.1537/I > 2\sigma(I)$ [...] are undoubted.” This statement is definitely false, as $R1$ above 0.1 already renders the structure unreliable. Moreover, CHECKCIF clearly recognizes crystal twinning, which should be treated systematically, instead of data removal.
- 3) Page 4, lines 69-70: “Crystals of the lcp phase (1-lcp) were obtained by stabilizing the as-grown crystals in air for 3 h”. What is the reason for this 3-hour long stabilization period? This may result in partial exchange of crystallization solvent (compound crystallizes from MeOH/H₂O mixture) and/or crystallization solvent loss. Both factors may contribute to the bad quality of single-crystal X-ray diffraction data for 1-lcp. The authors should repeat diffraction experiments for the crystal as-grown from the reaction mixture.
- 4) Page 5, line 80: “The crystals of the nqp phase 1·4/3H₂O were obtained by heating 1·9/2H₂O to 433 K for 48 h under vacuum”. Very long heating at 433 K may lead to crystal decomposition, as evidenced by PXRD pattern of 1 after heating at 433 K for 48h under vacuum (Figure 4 in the SI), which shows very large peak broadening and low signal/noise ratio. Therefore it is mandatory to perform TGA experiment at low heating rate (<1 K/min) for 1·9/2H₂O, in order to determine the lowest possible temperature that will facilitate water loss. Authors seem to have access to the necessary equipment, as they have demonstrated TGA results in their previous papers (for example Inorg. Chem. 2021, 60, 7337-7344).
- 5) Page 5, lines 80-81 and lines 92-93: “The crystals of the nqp phase 1·4/3H₂O were obtained by heating 1·9/2H₂O to 433 K for 48 h under vacuum and cooling in air” and “Notably, an attempt to obtain the crystal structure of completely dehydrated crystal of 1 failed because it was instantly converted to 1·4/3H₂O”. Cooling in humid air may result in partial rehydration, but in spite of that authors should characterize crystal structure of anhydrous 1. This should be achieved by gentle heating of 1·9/2H₂O in dry nitrogen stream in situ before the single-crystal diffraction measurement (the exact conditions may be deduced from TGA).
- 6) Water adsorption isotherm depicted in the Figure 2A may be influenced by partial sample decomposition, resulting from very long activation at high temperature (please compare with Figure 4 in the SI). The identity of the sample should be re-tested by PXRD measurement after water sorption experiment.
- 7) 1·4/3H₂O line in the Figure 2C actually corresponds to 5/3H₂O level, which is misleading to the reader and should be corrected. Moreover the “Uptake” drops down to -0.2 mol/mol for 1200 min., which may be another sign of partial sample decomposition at 450 K and undermines validity of this plot.
- 8) Authors provide long discussion of PXRD results in the attempt to prove the phase-coexistence in the single crystal. I need to emphasize that it is impossible to draw conclusions about single crystal on

the basis of powder measurements. There are many factors that may be responsible for observed changes in PXRD pattern, phase-coexistence within single crystal is not necessarily one of them.

9) Page 7, lines 148-150: “the phase-coexistence state works on individual crystals rather than a physical mixture of two pure phases because the PXRD patterns of a physical mixture of the lcp phase 1.9/2H₂O and nqp phase 1.4/3H₂O did not show any significant change after spraying the water mist Supplementary Fig. 14)”. This conclusion is in contradiction to the water adsorption isotherm (Figure 2B) which clearly shows that nqp-to-lcp transition appears above p/p₀ = 0.8. Lack of significant changes in the PXRD may simply result from limited vapor diffusion or too short stabilization time.

10) Page 7, lines 155-157: “the phase-coexistence state of individual crystal is directly supported by single-crystal diffraction analysis, the crystal lattices of both nqp and lcp phases can be indexed in reciprocal space from the data collected on a partially desolvated single crystal at 310 K”. It is not unusual to observe overlapping diffraction patterns in a single-crystal diffraction experiment. This may result from crystal breaking, twinning (which is most probably the case, as it is easily recognized by CHECKCIF in crystal structures of 1-nqp and 1-lcp) and other factors. It would require spatially-resolved diffraction with use of highly focused beam in order to prove phase-coexistence in the single crystal in a diffraction experiment (please see SI in J. Am. Chem. Soc. 2015, 137, 15390-15393 or Cryst. Eng. Comm. 2018, 20, 2233-2236 for similar experiments on welded crystals, where macroscopic domains were identified).

11) Magnetic measurements seem to contradict the hypothesis of phase-coexistence in single crystals of 1. Figures 3B-D look like linear combinations of curves presented in Figure 3A and Figure 3E/3F. This would be highly unexpected for the phase-coexistence within single crystal, as SCO in this system is highly cooperative. Moreover, small changes in water content may very strongly influence SCO behavior, even in the absence of phase-coexistence phenomenon (well depicted by Song et al. in Dalton Trans. 2016, 45, 18643-18652).

12) Sample preparation for magnetic measurements should be described in details in the discussion of magnetic properties (exact conditions of sample stabilization that lead in preparation of each phase mixture).

13) Methods section should contain description of the measurement setup (for HTK 1200N and TTK450) as well as sample preparation for PXRD measurements. Type of the sample holder and sample thickness are critical for assessing efficiency of vapor diffusion in such experiments.

14) Methods section should contain detailed description of sample packing for magnetic measurements. How was the sample stability achieved?

There are also several minor issues that should be corrected:

1) Page 2, line 16: “...leads to a pore rearrangement associated with local negative adsorption”. No negative adsorption is observed for the presented network and therefore this part of the sentence is an unnecessary exaggeration. The observed behavior is pore rearrangement only and the new term “local negative adsorption” should not be coined.

2) The first paragraph of the introduction section is not clear and especially lines 30-40 need rephrasing.

3) Page 3, line 31: “extend-retract motion¹²⁻¹⁴”. References 12-14 are not a good demonstration of this term.

4) Page 4, line 75: “3D porous structure with channel pores (15.027 × 6.554 Å²)”. Even assuming that atoms are dimensionless, the pore dimensions would be 14.3 Å (for Pt-Pt) × 5.3 Å (for H-H of neighboring organic ligands). The authors should clearly specify which distances were listed as “channel pores” dimensions.

5) On numerous occasions distances in the crystal structure are mentioned without uncertainty of their determination, which should be corrected.

6) CHECKCIF report for 1-nqp at 100K contains the following alert: PLAT920_ALERT_1_B Theta(Max) in CIF and FCF Differ by 2.68 Degree. Please clarify what is the reason for that.

7) Kepert et al. reported Hofmann-type frameworks based on very similar ligands and showing almost identical topology as 1-nqp (Chem. Sci. 2018, 9, 5623-5629 and Dalton Trans. 2021, 50, 1434-1442) – those references should be mentioned in the introduction.

8) I lack the expertise to evaluate details of DFT calculations, but I was alerted by the following description: "The experiment obtained two crystal structures [...] the other is lcp phase structure with 10 water molecules in the unit cell, denoted as lcp-10w". Why 10 water molecules were assumed in DFT calculations, whereas there are 9 water molecules in the crystal structure of 1-lcp?

Reviewer #2 (Remarks to the Author):

This manuscript reports a new example of an iron/palladium 2D Hoffmann network structure with 1D porosity. Such compounds have been heavily studied by the spin-crossover materials community for the past 20 years. They are well-known to afford interesting and useful thermal spin-transitions, which may be coupled to the removal or exchange of guest molecules within their pore structure. This is another example. It can be isolated in two different hydration states, which can be interconverted in single-crystal-to-single-crystal fashion. The two forms have very different pore structures and must interconvert via a detectable, metastable intermediate phase. That leads to gating during the interconversion of the two phases, which allows the mechanism of the interconversion to be monitored in unusual detail.

The design of the material is unexceptional, and single-crystal-to-single-crystal reactions in spin-crossover crystals are also quite well known now. Rather, the main interest lies in the complicated mechanism of the pore rearrangement between the phases, and how this manifests in the spin-transitions of the different phases. This has been thoroughly characterized by diffraction, by adsorption isotherms, and by periodic DFT calculations.

The authors have done a lot of experiments to sort this out. Some data aren't of the highest quality, but it's clear their interpretation and conclusions are sound. This is also novel enough for the journal, and it can be published if the following comments are addressed:

(i) The microanalyses of the two phases are good. However, a straightforward check would be to run thermogravimetric analyses (TGAs) on the two phases, as an extra confirmation of their water content. I was surprised those weren't included in the SI, and they should be added to the revision.

(ii) The crystal structures of the 1.nqp phase are of low quality. The 100 K structure is publishable, but the 250 K structure has very low data completeness, and too few observed data for a meaningful refinement – the observed data:parameter ratio is only 6.5:1. That explains the very low precision of that structure (Table 5, supporting information). There is also obvious disorder in the prentz ligands at that temperature, which hasn't been modelled; in fact, it can't be modelled with such a weak dataset.

The basic interpretation of that structure is doubtless correct, but I wouldn't publish such a low quality crystal structure without a good reason. In fact, the 100 K structure gives all the information needed to interpret the magnetic data of 1.nqp, so my choice would be to remove the 250 K structure from the manuscript.

In any case, the authors should comment on the low quality diffraction shown by 1.nqp – it's not truly a single-crystal-to-single-crystal transformation if the crystal degrades badly during the process.

(iii) Why is the spin-transition at Fe(2) in 1.nqp incomplete? It probably isn't kinetically trapped, as the transition temperature is too high for that.

(iv) It looks like there is strong preferred orientation in the powder diffraction data in Figures 7-10 of the SI. That doesn't invalidate the measurements but it should be explained – were those samples true powders, or crystals?

(v) Figures 7-10 are also so compressed, that it's hard to see what's going on. I would expand them vertically, and just show one Figure per page, so we can see the powder patterns properly.

(vi) The text says Rietveld refinements were obtained for both phases (line 96), but only 1.lcp is

included in the SI (Figure 18). The Rietveld refinement for 1.nqp should be added to the revision.

(vii) I know of one other spin-crossover material, where different phases coexist in the same crystal during a desolvation process (Chem. Sci. 2016, 7, 2907). That should be cited in the revision.

(viii) 'lcp' and 'nqp' are non-standard abbreviations. They are defined in the Abstract, but they should also be defined in the main text so the reader easily understands what they mean.

Responses to reviewers' comments

We are grateful to all reviewers for their thorough evaluation of our manuscript entitled "Single crystal-to-single crystal and spin-crossover transformations via phase-coexistence state in pore-adjustable frameworks" (NCOMMS-21-40232). Their kind comments and suggestions would highly help us improve the quality of our manuscript. Now, we have done additional experiments and addressed all questions raised by the reviewers. Changes made in the manuscript and supplementary material have been highlighted in yellow.

Our point-by-point responses to all reviewers' comments are listed below, shown in italic and red.

Responses to reviewer #1:

Comments to Authors:

The authors describe a gate-opening adsorption in the Hofmann-type framework, which demonstrates water-induced variations of spin crossover. This is interesting on its own, but was previously presented for a Hofmann-type framework by Real et al. (*Chem. Sci.* **11**, 11224-11234 (2020); reference 40). Moreover, authors postulate presence of the "phase-coexistence state of individual crystals". Such an observation would be very valuable, but in my opinion authors do not present sufficient scientific evidence to support this claim. The design of the appropriate experiment is challenging, but it seems clear that it would require use of space-resolved techniques, while the authors performed most of the characterization on bulk powder samples (in addition to the reference 47, I encourage the authors to read the article by Neimark *et al.* in *J. Phys. Chem. Lett.* **2**, 2033-2037 (2011).). Moreover, manuscript contains numerous technical drawbacks that must be resolved before publication:

Response: Thank you for evaluating our manuscript and providing your meticulous and very valuable comments to help us improve this manuscript to a much better scientific level. As you suggested, phase-coexistence state of individual crystals is very valuable, particularly when it is associated with a porous structural transformation as it may lead to intriguing adsorption properties that have not been identified in previous study. To address your concerns on the existence of phase-coexistence state of individual crystals, we performed more experiments, including better single crystal data, TG measurements, water adsorption isotherms measurements, PXRD, and micro-Raman spectra measurements, and also some sentences and

Figures were modified according to your suggestions. We believed these new evidences are sufficient to support our claim.

1. Single-crystal X-ray diffraction data for 1-lcp phase lead to structure solutions that show very large residual electron density in the proximity of the palladium ion. Authors use hundreds of OMIT instructions (more than 700 in case of 1-lcp at 150 K) in order to artificially remove this electron density, which is a case of data manipulation that should never be performed. Instead, an adequate twin law should be applied to systematically remove electron density attributed to the twinned crystal or the experiment itself should be repeated (comments 3 and 5).

Response: According to your suggestion, we performed further refinements on the single-crystal structure of 1-lcp (1.9/2H₂O) using the twin law of CrysAlisPro XtaLAB PRO system (Twinning-multi-crystals), and refinement results were remarkably improved. For the crystal structures at 85 and 150 K, the largest residual electron densities were reduced to values of 1.0 and 2.3 eÅ⁻³, respectively. However, the largest residual

electron density of $4.74 \text{ e}\text{\AA}^{-3}$ in the vicinity of palladium ion remained at 250 K (B-level alert). To completely address your concerns, we further measured the single-crystal structure for **1-1cp** at 293 K using a single crystal with smaller size at the BL17B beam line station of the Shanghai Synchrotron Radiation Facility (SSRF). This refinement result is high quality without large residual electron density ($1.9 \text{ e}\text{\AA}^{-3}$).

The X-ray crystallographic coordinates for structures have been updated and deposited at the Cambridge Crystallographic Data Center (CCDC). The slight changes in the bond lengths were also updated in the manuscript and supplementary table as below:

In the manuscript:

Page 7-8: The average Fe1–N bond lengths are $2.167(5) \text{ \AA}$ at 250 K and $1.964(5) \text{ \AA}$ at 150 K, respectively, indicating that a complete HS-LS transition of Fe1 occurs in the first step spin transition upon cooling (Supplementary Figs. 16–18 and Table 4). The average Fe2–N bond length shortens from $2.160(4) \text{ \AA}$ at 150 K to $1.970(4) \text{ \AA}$ at 85 K, showing that Fe2 undergoes a complete spin transition at the lower temperature range.

Page 10: This is particular in the present case because the excluded water molecules exhibit strong hydrogen bonds with the N atoms of the framework with O...N distances of $2.754(2) \text{ \AA}$ and $2.908(3) \text{ \AA}$ (250 K, Fig. 4c and Supplementary Fig. 23 and Table 6).

In the single-crystal X-ray diffraction (SC-XRD) analyses:

Page 13: The crystal data of $1\cdot 9/2\text{H}_2\text{O}$ and $1\cdot 4/3\text{H}_2\text{O}$ was further collected with a Bruker/ARINAX MD2 diffractometer equipped with a MarCCD-300 detector ($\lambda = 0.77484 \text{ \AA}$) at the BL17B beam line station of the Shanghai Synchrotron Radiation Facility (SSRF). The data of $1\cdot 9/2\text{H}_2\text{O}$ was collected at room temperature (293 K) and the data of $1\cdot 4/3\text{H}_2\text{O}$ was collected at 100 K. 360 frames of each data were collected using ω -scans with oscillation angle of 1° , scan range of 0 to 360° , exposure time of 0.50 s per frame, and detector distance of 90 mm.

In the data availability:

Page15: All data in this study are presented in the article and its Supplementary Information, which are also available from the authors upon request. The X-ray crystallographic coordinates for structures have been deposited at the Cambridge Crystallographic Data Center (CCDC) under deposition numbers CCDC 2093900 ($1\cdot 4/3\text{H}_2\text{O}$ at 100 K), 2111045 ($1\cdot 4/3\text{H}_2\text{O}$ at 250 K), 2124267 ($1\cdot 9/2\text{H}_2\text{O}$ at 293 K), 2093901 ($1\cdot 9/2\text{H}_2\text{O}$ at 250 K), 2093902 ($1\cdot 9/2\text{H}_2\text{O}$ at 150 K), and 2093903 ($1\cdot 9/2\text{H}_2\text{O}$ at 85 K). These data can be obtained free of charge via http://www.ccdc.cam.ac.uk/data_request/cif.

In the SI:

Supplementary Table 1 Crystal data and structural refinements for $1\cdot 9/2\text{H}_2\text{O}$.

	1·9/2H ₂ O			
Temperature/K	293	250	150	85
Formula	C ₅₂ H ₅₈ Fe ₂ N ₂₄ O ₉ Pd ₂			
Mr/g mol ⁻¹	1487.72			
Crystal size/mm	0.05×0.08×0.03		0.1×0.15×0.03	
Wavelength/Å	0.77484		0.71073	
Space group	P $\bar{1}$	P $\bar{1}$	P $\bar{1}$	P $\bar{1}$
Crystal color	yellow	yellow	red	red
Crystal system	triclinic	triclinic	triclinic	triclinic
a/Å	7.4570(15)	7.4527(3)	7.2818(2)	7.1967(2)

b /Å	14.711(3)	14.7570(7)	14.5355(3)	14.2609(4)
c /Å	15.442(3)	15.4109(10)	15.3506(5)	15.2514(6)
α /°	104.49(3)	104.331(5)	104.332(2)	103.854(3)
β /°	99.92(3)	99.942(4)	100.229(2)	99.891(3)
γ /°	90.28(3)	90.138(4)	90.260(2)	90.132(2)
Volume/Å ³	1613.5(6)	1615.67(15)	1547.18(8)	1495.60(9)
Z	1	1	1	1
D _c / g cm ⁻³	1.531	1.529	1.597	1.652
μ / mm ⁻¹	1.060	1.058	1.105	1.143
F (000)	754.0	754.0	754.0	754.0
$\square\square$ range/°	1.43 to 25.0	3.89 to 25.03	3.50 to 30.00	3.53 to 29.92
Data/restraints/parameters	5185/18/418	5484/37/419	22636/6/419	21087/2/422
Goodness-of-fit on F ²	1.061	1.057	1.063	1.042
Reflections collected	10024	5684	22636	21087
R ₁ [I ≥ 2σ(I)]	0.0631	0.0731	0.0570	0.0466
wR ₂ [all data]	0.1826	0.1943	0.1897	0.1341
Largest diff. peak and hole/e.Å ⁻³	1.89/-1.41	4.74/-3.20	2.25/-1.30	1.04/-0.97
Completeness	91.5%	99.6%	99.6%	99.7%

Supplementary Table 4 Selected bond lengths and angles for 1·9/2H₂O at different temperatures.

	85 K	150 K	250 K	293 K
Fe1–N1	1.957(4)	1.957(5)	2.161(5)	2.154(5)
Fe1–N2	1.957(4)	1.952(4)	2.157(5)	2.159(5)
Fe1–N3	1.975(3)	1.982(4)	2.181(5)	2.164(5)
Fe2–N7	1.962(4)	2.145(4)	2.168(5)	2.152(5)
Fe2–N8	1.960(4)	2.160(5)	2.156(5)	2.171(5)
Fe2–N9	1.989(3)	2.175(4)	2.196(5)	2.185(5)
C1–N1–Fe1	178.5(4)	177.9(4)	178.9(6)	177.3(5)
C2–N2–Fe1	179.9(5)	179.4(5)	177.9(6)	178.7(5)
C3–N3–Fe1	129.2(3)	129.4(4)	129.2(3)	129.8(4)
C14–N7–Fe2	173.9(4)	163.4(4)	169.5(5)	163.8(5)
C15–N8–Fe2	171.3(4)	170.4(5)	164.9(5)	169.3(5)
C16–N9–Fe2	128.3(3)	125.6(4)	126.0(5)	126.4(4)

Supplementary Table 6 Selected structural parameters for 1·9/2H₂O and 1·4/3H₂O at different temperatures.

Compound	1·9/2H ₂ O				1·4/3H ₂ O	
	85 K	150 K	250 K	293 K	100 K	250 K
<Fe1–N> ^[a] /Å	1.963(4)	1.964(4)	2.167(5)	2.159(5)	1.949(7)	2.134(9)
<Fe2–N> ^[a] /Å	1.970(4)	2.160(4)	2.173(5)	2.169(5)	2.062(3)	2.144(9)
ΣFe1 ^[b]	11.24(14)	8.28(18)	14.0(2)	12.92(18)	6.4(15)	13.2(6)
ΣFe2 ^[b]	9.24(15)	14.04(17)	7.2(2)	5.12(18)	19.4(11)	11.5(4)

$d_{N-O}^{[c]}$	2.839(4)	2.847(4)	2.908(3)	2.896(4)	2.798(3)	2.754(2)
					2.940(4)	3.018(5)
$\langle Fe1-N-C \rangle^{[d]}$	179.20(5)	178.60(5)	178.41(3)	178.01(5)	177.4(5)	174.25(5)
$\langle Fe2-N-C \rangle^{[d]}$	172.56(2)	166.93(5)	167.26(3)	166.62(4)	175.35(3)	173.90(3)
$\Sigma \tau^{[e]}/^\circ$	4.72(6)	17.20(8)	7.46(5)	7.80(3)	5.043(12)	8.05(13)

[a] The average Fe–N bond lengths (Å);

[b] Octahedral distortion parameters (°);

[c] The distance of hydrogen bonds between uncoordinated nitrogen atom of 1,2,4-triazole in the prentz ligand and oxygen atom of adjacent water;

[d] Average Fe–N–C angles within Hofmann layer;

[e] The parameter is defined as the average of the sum of $|90^\circ - \theta|$ for Pd–Fe–Pd and Fe–Pd–Fe angles within the Hofmann layer (Three Pd–Fe1–Pd, three Pd–Fe2–Pd, two Fe1–Pd–Fe1, two Fe2–Pd–Fe2 and two Fe2–Pd–Fe1 angles of 1·9/2H₂O; Three Pd–Fe1–Pd, five Pd–Fe2–Pd, two Fe1–Pd–Fe1, four Fe2–Pd–Fe2 and two Fe2–Pd–Fe1 angles of 1·4/3H₂O).

2. Crystal structure 1-nqp at 250 K is characterized by $R_1 > 0.15$ and data completeness below 73%. In the CIF file authors claim: “The final refinement results with low R_1 : 0.1537// $> 2\sigma(I)$ [...] are undoubted.” This statement is definitely false, as R_1 above 0.1 already renders the structure unreliable. Moreover, CHECKCIF clearly recognizes crystal twinning, which should be treated systematically, instead of data removal.

Response: We remeasured the crystal structure of **1-nqp** (1·4/3H₂O) at 250 K with a high-quality single crystal, and the structural refinement were significantly improved with $R_1 = 0.0984$ and data completeness = 99.7%. This new result has been updated in CCDC and the manuscript as below, and the claim in the CIF file has been removed.

In the manuscript:

Page 8: The average Fe1–N bond lengths of 2.134(9) Å at 250 K and 1.949(6) Å at 100 K (Supplementary Table 5), respectively, suggest that Fe1 ion in 1·4/3H₂O undergoes an complete SCO. While for Fe2 ion, an incomplete SCO is concluded as the average Fe2–N bond lengths shorten from 2.144(10) Å at 250 K to 2.062(7) Å at 100 K.

Page 10: This is particular in the present case because the excluded water molecules exhibit strong hydrogen bonds with the N atoms of the framework with O...N distances of 2.754(2) Å and 2.908(3) Å (250 K, Fig. 4C and Supplementary Fig. 23 and Table 6).

In the SI:

Supplementary Table 2 Crystal data and structural refinements for 1·4/3H₂O.

	1·4/3H ₂ O	
Temperature/K	100	250
Formula	C ₇₈ H ₆₈ Fe ₃ N ₃₆ O ₄ Pd ₃	
$M_r/g\ mol^{-1}$	2060.43	
Crystal size/mm	0.08×0.07×0.03	0.18×0.1×0.03
Wavelength/Å	0.77484	0.71073

Space group	$P\bar{1}$	$P\bar{1}$
Crystal color	red	yellow
Crystal system	triclinic	triclinic
a/Å	7.2100(14)	7.4258(5)
b/Å	16.097(3)	16.3023(11)
c/Å	19.051(4)	19.2325(12)
α /°	76.31(3)	76.768(6)
β /°	80.61(3)	80.084(6)
γ /°	79.15(3)	78.605(6)
Volume/Å ³	2093.5(8)	2202.0(3)
Z	1	1
D_c / g cm ⁻³	1.634	1.554
μ / mm ⁻¹	1.210	1.150
F(000)	1036.0	1036.0
χ^2 range/°	1.318 to 27.09	3.511 to 26.371
Data/restraints/parameters	7277/6/564	13208/18/500
Goodness-of-fit on F^2	1.129	1.032
Reflections collected	33449	13208
R_1 [$I \geq 2\sigma(I)$]	0.0881	0.0984
w R_2 [all data]	0.2671	0.2737
Largest diff. peak and hole/e.Å ⁻³	2.17/-2.33	4.94/-1.63
Completeness	89.0%	99.7%

Supplementary Table 5 Selected bond lengths and angles for 1·4/3H₂O.

	100K	250 K
Fe1–N1	1.940(5)	2.134(10)
Fe1–N2	1.941(6)	2.129(10)
Fe1–N7	1.965(6)	2.138(6)
Fe2–N3	2.059(5)	2.157(17)
Fe2–N4	2.039(7)	2.132(10)
Fe2–N5	2.052(8)	2.145(10)
Fe2–N6	2.061(7)	2.169(10)
Fe2–N11	2.096(8)	2.167(7)
Fe2–N15	2.062(6)	2.105(7)
C1–N1–Fe1	177.3(6)	173.2(11)
C2–N2–Fe1	177.5(6)	175.4(11)
C7–N7–Fe1	130.0(5)	128.1(5)
C3–N3–Fe2	175.3(6)	173.7(11)
C4–N4–Fe2	170.1(6)	168.7(11)
C5–N5–Fe2	176.5(7)	175.6(12)
C6–N6–Fe2	179.5(8)	177.7(11)
C18–N11–Fe2	123.1(6)	118.4(5)
C29–N15–Fe2	126.3(4)	126.2(5)

Supplementary Table 6 Selected structural parameters for 1·9/2H₂O and 1·4/3H₂O at different

temperatures.

Compound	1·9/2H ₂ O				1·4/3H ₂ O	
	85 K	150 K	250 K	293 K	100 K	250 K
<Fe1–N> ^[a] /Å	1.963(4)	1.964(4)	2.167(5)	2.159(5)	1.949(7)	2.134(9)
<Fe2–N> ^[a] /Å	1.970(4)	2.160(4)	2.173(5)	2.169(5)	2.062(3)	2.144(9)
∑Fe1 ^[b]	11.24(14)	8.28(18)	14.0(2)	12.92(18)	6.4(15)	13.2(6)
∑Fe2 ^[b]	9.24(15)	14.04(17)	7.2(2)	5.12(18)	19.4(11)	11.5(4)
d _{N–O} ^[c]	2.839(4)	2.847(4)	2.908(3)	2.896(4)	2.798(3)	2.754(2)
					2.940(4)	3.018(5)
<Fe1–N–C> ^[d]	179.20(5)	178.60(5)	178.41(3)	178.01(5)	177.4(5)	174.25(5)
<Fe2–N–C> ^[d]	172.56(2)	166.93(5)	167.26(3)	166.62(4)	175.35(3)	173.90(3)
∑ ^[e] /°	4.72(6)	17.20(8)	7.46(5)	7.80(3)	5.043(12)	8.05(13)

[a] The average Fe–N bond lengths (Å);

[b] Octahedral distortion parameters (°);

[c] The distance of hydrogen bonds between uncoordinated nitrogen atom of 1,2,4-triazole in the prentzr ligand and oxygen atom of adjacent water;

[d] Average Fe–N–C angles within Hofmann layer;

[e] The parameter is defined as the average of the sum of $|90^\circ - \theta|$ for Pd–Fe–Pd and Fe–Pd–Fe angles within the Hofmann layer (Three Pd–Fe1–Pd, three Pd–Fe2–Pd, two Fe1–Pd–Fe1, two Fe2–Pd–Fe2 and two Fe2–Pd–Fe1 angles of 1·9/2H₂O; Three Pd–Fe1–Pd, five Pd–Fe2–Pd, two Fe1–Pd–Fe1, four Fe2–Pd–Fe2 and two Fe2–Pd–Fe1 angles of 1·4/3H₂O).

3. Page 4, lines 69-70: “Crystals of the lcp phase (1-lcp) were obtained by stabilizing the as-grown crystals in air for 3 h”. What is the reason for this 3-hour long stabilization period? This may result in partial exchange of crystallization solvent (compound crystallizes from MeOH/H₂O mixture) and/or crystallization solvent loss. Both factors may contribute to the bad quality of single-crystal X-ray diffraction data for 1-lcp. The authors should repeat diffraction experiments for the crystal as-grown from the reaction mixture.

Response: The pristine crystals prepared from MeOH/H₂O mixture contain both H₂O and MeOH molecules (see Table R1 and Fig. R1a) and its SCO property is different from the single crystal of 1·9/2H₂O (Fig. R1b). Our primary experiment shows the spin equilibrium of the pristine single crystal can be affected by the methanol molecules, and a further experimental and theoretical study is in progress to investigate the intercorrelation between the molecular dynamic and spin equilibrium. As the quality of single-crystal structural refinements has been significantly improved and the structure of pristine crystals is independent from the conclusion of this manuscript, it would be better that the structure of pristine crystals and detailed magnetic property can be presented together with a thorough mechanistic explanation in a future study.

As you suggested, the pristine crystals should involve in a partial exchange of crystallization solvent when they are exposed in the air. The fast exchange and removal of MeOH guests were supported by TG measurement. As shown in TG curve below, the MeOH guests and surface solvent of the pristine samples can be easily removed, while the residual H₂O molecules (9/2H₂O) in the cavity is stable in the room temperature. The stabilization time of 3 h in the manuscript is the procedure we used to get the pure

1·9/2H₂O single crystal. To clarify this point, the sentence in the manuscript has been changed and the result of TG measurement was added in the SI as Supplementary Fig. 1.

Fig. R1 The structure and magnetic property of pristine crystals (1·xMeOH·yH₂O). **A**, the asymmetric unit of the pristine crystal. **B**, Temperature dependence of the $\chi_M T$ curves of the pristine crystals.

Table R1 Crystal data and structural refinements for 1·xMeOH·yH₂O.

1·xMeOH·yH ₂ O	
Temperature/K	150
Crystal size/mm	0.07×0.06×0.03
Wavelength/Å	0.71073
Space group	$P\bar{1}$
Crystal color	red
Crystal system	triclinic
a/Å	7.2770(15)
b/Å	14.509(3)
c/Å	15.381(3)
α /°	104.67(3)
β /°	99.99(3)
γ /°	90.32(3)
Volume/Å ³	1545.1(6)
Z	1
D_c / g cm ⁻³	1.539
μ / mm ⁻¹	1.103
F(000)	698
ω range/°	1.391 to 27.13
Data/restraints/parameters	5955/0/430
Goodness-of-fit on F^2	1.055
Reflections collected	11173
R_1 [$I \geq 2\sigma(I)$]	0.0690
wR_2 [all data]	0.1963
Largest diff. peak and hole/e.Å ⁻³	2.13/-2.14

In the manuscript:

Page 3: Crystals of the lcp phase (1·9/2H₂O) were obtained by stabilizing the as-grown crystals that contained H₂O and MeOH mixture guests in air for 3 h (Supplementary Fig. 1).

In the SI:

Supplementary Fig. 1 Thermogravimetric analysis of the pristine crystal. The pristine crystals prepared from MeOH/H₂O mixture contain both H₂O and MeOH molecules that can be partially exchanged when the sample is exposed in the air. The first step weight loss in the TG curve suggests a fast exchange and removal of MeOH guests in the room temperature, and the second step weight loss corresponds to the 1·9/2H₂O to 1·4/3H₂O transformation, and the third step weight loss indicates the 1·4/3H₂O to 1 transformation. Notably, although the TG measurement suggests the guest H₂O molecules were removed at ca. 400 K, a small part of 1·9/2H₂O (lcp phase structure) remains according to PXRD and vapor adsorption isotherm measurements.

In the SI Method:

Page S5: Thermogravimetric analysis (TGA) of the pristine crystal was recorded under a nitrogen flow (200 mL min⁻¹) on a Shimadzu DTG-60AH instrument in a temperature range of 293–775 K with rates of 0.25 K min⁻¹ for 293 to 423 K, 0.5 K min⁻¹ for 423 to 473 K, and 2 K min⁻¹ for 473 to 773 K.

4. Page 5, line 80: “The crystals of the nqp phase 1·4/3H₂O were obtained by heating 1·9/2H₂O to 433 K for 48 h under vacuum”. Very long heating at 433 K may lead to crystal decomposition, as evidenced by PXRD pattern of 1 after heating at 433 K for 48h under vacuum (Figure 4 in the SI), which shows very large peak broadening and low signal/noise ratio. Therefore it is mandatory to perform TGA experiment at low heating rate (<1 K/min) for 1·9/2H₂O, in order to determine the lowest possible temperature that will facilitate water loss. Authors seem to have access to the necessary equipment, as they have demonstrated TGA results in their previous papers (for example *Inorg. Chem.* **60**, 7337-7344 (2021)).

Response: We performed TGA experiments of the lcp phase 1·9/2H₂O and nqp phase 1·4/3H₂O in a temperature range of 293–775 K with rates of 0.25 K min⁻¹ for 293 to 423 K, 0.5 K min⁻¹ for 423 to 473 K, and 2 K min⁻¹ for 473 to 773 K under a N₂ flow. The new results are shown below and Supplementary Figure

5. The TG measurement reveals that lcp phase 1·9/2H₂O undergoes a two-step weight loss (one step for 1·4/3H₂O) upon heating, and the total weight loss of 10.9% indicates that almost all of water molecules are removed at ca. 375 K. However, both the PXRD measurements and adsorption isotherm reveal that the pure nqp phase cannot be prepared even the sample was heated at 433 K for 18 h. In fact, the abnormal robust survivability of lcp phase further supports our mechanism that pore rearrangement associates with inner water redistribution during adsorption, as a trace amount of water could be locked in the pores and it impedes

the local lcp to nqp structural transformation.

As both the TG and PXRD measurements indicate the water-free sample is stable up to 503 K, the pure crystals of the nqp phase were obtained by heating $1\cdot9/2\text{H}_2\text{O}$ to 433 K for 48 h under vacuum to reduce the preparing time. Notably, the high-quality single crystals of $1\cdot4/3\text{H}_2\text{O}$ prepared by heating $1\cdot9/2\text{H}_2\text{O}$ at 433 K for 48 h under vacuum and stored in the dry air for 3 months is capable of single-crystal X-ray diffraction analyses, further supporting the good stability of crystal lattice.

We remeasured the PXRD pattern of **1** after heating the dried sample at 433 K for 48 h under vacuum (the heating rate of 5 K min^{-1}). The peak broadening and signal/noise ratio of the PXRD pattern have been improved.

These new results have been added in the Supplementary Information:

Supplementary Fig. 5 Thermogravimetric analysis of lcp phase $1\cdot4/3\text{H}_2\text{O}$ (A) and nqp phase $1\cdot4/3\text{H}_2\text{O}$ (B). The anhydrous sample is stable up to ca. 503 K.

Supplementary Fig. 8 PXRD patterns of **1** upon heating the dried sample at 433 K for 48h under vacuum (the heating rate of 5 K min^{-1}). The PXRD data of **1** was collected at room temperature after activated treatment.

- Page 5, lines 80-81 and lines 92-93: "The crystals of the nqp phase $1\cdot4/3\text{H}_2\text{O}$ were obtained by heating $1\cdot9/2\text{H}_2\text{O}$ to 433 K for 48 h under vacuum and cooling in air" and "Notably, an attempt to obtain the crystal structure of completely dehydrated crystal of **1** failed because it was instantly converted to $1\cdot4/3\text{H}_2\text{O}$ ". Cooling in humid air may result in partial rehydration, but in spite of that authors should

characterize crystal structure of anhydrous **1**. This should be achieved by gentle heating of $1 \cdot 9/2\text{H}_2\text{O}$ in dry nitrogen stream *in situ* before the single-crystal diffraction measurement (the exact conditions may be deduced from TGA).

Response: We indeed tried several times to carry out the single-crystal diffraction experiment of **1** by *in situ* heating the crystal at 375 or 390 K in dry N_2 stream, but the crystal structure of anhydrous crystal of **1** cannot be obtained because of the poor diffraction. But the decomposition of framework of **1** should be precluded, as we can re-obtain the crystal structure of $1 \cdot 4/3\text{H}_2\text{O}$ by exposing one of the above single crystals in the air.

To clarify this point, the sentence in the manuscript have revised as below:

Page 4: **Notably, an attempt to obtain the crystal structure of completely dehydrated crystal of **1** failed because of the poor diffraction of anhydrous crystal of **1**.**

6. Water adsorption isotherm depicted in the Figure 2A may be influenced by partial sample decomposition, resulting from very long activation at high temperature (please compare with Figure 4 in the SI). The identity of the sample should be re-tested by PXRD measurement after water sorption experiment.

Response: To verify the adsorption isotherm depicted in the Figure 2A is not induced by partial sample decomposition, we re-tested the sample by PXRD measurement after water sorption experiment. As shown in Figures below, the PXRD patterns of sample after two isothermal vapor ad/de-sorption experiments are

almost same, therefore, the large difference of water adsorption in the Figure 2a and 2b should not be induced by the sample decomposition. Moreover, the PXRD of sample also remains unchanged after the continuous adsorption-desorption-cycles measurement in the Fig. 2B.

The new result has been added in the SI as Supplementary Fig. 11.

Supplementary Fig. 11 PXRD patterns of samples before (Blue) and after (Green) two adsorption isotherm experiments and the continuous adsorption-desorption-cycles measurement. A, The PXRD pattern of sample after two isothermal vapor ad/de-sorption (in the Fig. 2A up) of the sample activated at 433 K under vacuum for 48 h (pure nqp phase). B, The PXRD pattern of sample after a continuous adsorption-desorption-cycles measurement (in the Fig. 2B, including a long-time heating at 433 K under vacuum and a mild condition at 303 K under vacuum) is same with that before sorption experiments, suggesting the robust crystal lattice of this sample.

7. $1 \cdot 4/3\text{H}_2\text{O}$ line in the Figure 2C actually corresponds to $5/3\text{H}_2\text{O}$ level, which is misleading to the reader and should be corrected. Moreover the "Uptake" drops down to -0.2 mol/mol for 1200 min., which may be another sign of partial sample decomposition at 450 K and undermines validity of this plot.

Response: We have corrected the misleading line in Figure 2C. The abnormal drop of "Uptake" line at 1200

min is ascribed to the change of heating source: Adsorption measurements are usually performed under a water bath with a temperature range of RT to 80°C, and desorption measurements are performed under a furnace with a temperature range of RT to 1000°C. As confirmed by repeating experiments (new results see Supplementary Fig.10 below), the switching of heating source is always accompanied with an anomaly in the “Uptake” curve.

Supplementary Fig. 10 The water adsorption-desorption measurements of the sample activated at different temperatures. (A), The sample activated at 400 K under vacuum for 24 h. (B), The sample activated at 433 K under vacuum for 24 h. Although the measurements indicate that almost all water guests are removed upon heating at 400 K, the uptake of ca. 2.6 H₂O per Fe^{II} for the sample activated at 400 K for 24 h is larger than that of ca. 1.7 H₂O per Fe^{II} for the sample activated at 433 K for 24 h, suggesting more crystals of phase-coexistence phase survived in the former, and this result is consistent with the PXRD measurements. Star marker: The abnormal drops of “Uptake” line is ascribed to the change of heating source

This information has been amended in the Figure legend of Figure 2 as below:

Fig. 2 Water adsorption isotherms and corresponding structural transformations probed by PXRD. **A**, Isothermal vapor adsorption and desorption of the sample activated at 433 K under vacuum for 48 h (pure nqp phase, up), the sample activated at 298 K under vacuum (phase-coexistence state, middle), and a physical mixture sample of 1·9/2H₂O and 1·4/3H₂O activated at 298 K under vacuum (down). The solid and open curves denote the vapor adsorption and desorption, respectively. **B**, The continuous water adsorption-desorption cycles in different activated conditions. The abnormal drop of “Uptake” line marked by star at 1200 min is ascribed to the change of heating source (Furnace and water bath). **C**,

PXRD patterns upon water adsorption and desorption. 1·9/2H₂O loses water rapidly upon heating (323 K), which leads to a phase-coexistence state that can return to the lcp phase immediately by spraying with a water mist. Such rapid water adsorption of the phase-coexistence sample was performed two times to

confirm the phenomenon. However, the completely dehydrated sample 1 prepared from long-time heating (433 K) under vacuum cannot recover to lcp phase smoothly upon water adsorption. In the measured sample, it took five days after leaving the pure 1 in saturated steam. **The PXRD patterns of the physical mixture of two pure phases did not show any significant change after spraying the water mist for 60 min.** The blue and red peaks denote the character peaks of structures of lcp and nqp phases, respectively.

8. Authors provide long discussion of PXRD results in the attempt to prove the phase-coexistence in the single crystal. I need to emphasize that it is impossible to draw conclusions about single crystal on the basis of powder measurements. There are many factors that may be responsible for observed changes in PXRD pattern, phase-coexistence within single crystal is not necessarily one of them. This comment is answered together with comment 10.
10. Page 7, lines 155-157: “the phase-coexistence state of individual crystal is directly supported by single-crystal diffraction analysis, the crystal lattices of both nqp and lcp phases can be indexed in reciprocal space from the data collected on a partially desolvated single crystal at 310 K”. It is not unusual to observe overlapping diffraction patterns in a single-crystal diffraction experiment. This may result from crystal breaking, twinning (which is most probably the case, as it is easily recognized by CHECKCIF in crystal structures of 1-nqp and 1-lcp) and other factors. It would require spatially-resolved diffraction with use of highly focused beam in order to prove phase-coexistence in the single crystal in a diffraction experiment (please see SI in *J. Am. Chem. Soc.* **137**, 15390-15393 (2015) or *Cryst. Eng. Comm.* **20**, 2233-2236 (2018) for similar experiments on welded crystals, where macroscopic domains were identified).

Response: I think these two comments reflect your main concerns about the validity of phase coexistence of the single crystal, so we combined the above two comments and our responses together. Indeed, drawing a conclusion about single crystal on the basis of only powder measurements is impossible. Hence, we performed the single-crystal X-ray diffraction on a phase-coexistence sample. As you pointed out in comment 10, overlapping diffraction patterns in a single-crystal diffraction experiment is usually own to the crystal breaking and twinning. However, the crystal domains in our sample have distinct cell parameters that denote to the crystal structures of nqp and lcp phases, respectively. As you know, this is different from typical crystal twinning in which the crystal domains usually possess same cell parameters.

Moreover, the overlapping diffraction patterns could be induced by the crystal breaking. In this case, the crystal can be regard as a physical mixture of two crystals of lcp and nqp phases. In our study, both the single-crystal X-ray diffraction, water adsorption isotherm, and PXRD measurements reveal the physical-mixture sample cannot recover to the lcp phase smoothly. However, when the single crystal of phase-coexistence state was exposed to the water, the lattice immediately recovered to the lcp phase within 30 min (see Figure below). This phenomenon is in contrast to a recovery time of 4 days required for the single crystal of pure nqp phase to lcp phase performed in the same condition. Therefore, the diffraction patterns

in the measured sample should be ascribed to the phase-coexistence state of individual crystal.

As you suggested, the spatially-resolved diffraction with use of highly focused beam is a powerful method to confirm the phase-coexistence in the single crystal. However, our single crystal is too thin and small to carry out the spatially-resolved diffraction. In addition to the microfocus X-ray diffraction, micro-Raman spectroscopy is also effective techniques to rationalize the distinction in different regions of a single crystal (*Nat. Chem.* 2015 **7**, 65-72; *Angew. Chem. Int. Ed.* 2017 **56**, 1-7). In order to present more evidence for the phase-coexistence of individual crystals, the micro-Raman spectroscopy experiments (spot diameter ~2 μm) were performed on single crystals.

In the micro-Raman spectra, the peaks at 1160 and 1178 cm^{-1} denote the structure of nqp phase and a

single broad peak at 1159 cm^{-1} denote the structure of lcp phase. In the pure nqp single crystal, the peaks at 1160 and 1178 cm^{-1} persisted up to 3 days in saturated steam, and then converted to lcp structure. While for a single crystal in phase-coexistence state, the peaks that denote to nqp structure can change to a single broad peak within 10 min in a humidity of 80%. The distinct responses of pure nqp single crystal and phase-coexistence single crystal in the micro-Raman spectra verify the phase-coexistence dependent water adsorption property in this material.

These results have been added in the manuscript and SI as below:

In the manuscript:

Page 6: **Phase-coexistence state investigated by SC-XRD and micro-Raman spectroscopy analyses.** The phase-coexistence state of individual crystal is directly supported by single-crystal diffraction analyses. The crystal lattices of both nqp and lcp phases can be indexed in reciprocal space from the data collected on a partially desolvated single crystal at 310 K (Supplementary Fig. 19 and Table 3), and this phase-coexistence state can immediately recovered to the lcp phase within 30 min upon exposing to the water. This phenomenon is in sharp contrast to a recovery time of 4 days required for a transformation of single crystal of pure nqp phase to lcp phase performed in the same condition, therefore precluding the crystal breaking of the phase-coexistence state.

Page 7: In order to verify the phase-coexistence of individual crystal, micro-Raman spectra experiments (spot diameter $\sim 2\text{ }\mu\text{m}$) were performed on single crystals^{56,57}. As shown in Supplementary Fig. 20, the peaks at 1160 and 1178 cm^{-1} denote the structure of nqp phase and a single broad peak at 1159 cm^{-1} denote the structure of lcp phase⁵⁸. In the pure nqp single crystal, the peaks at 1160 and 1178 cm^{-1} persisted up to 3 days in saturated steam, and then converted to lcp structure (Supplementary Fig. 21). While for a single crystal in phase-coexistence state, the peaks that denote to nqp structure can change to a single broad peak within 10 min in a humidity of 80% (Supplementary Fig. 22 and 23). The distinct responses of pure nqp single crystal and phase-coexistence single crystal in the micro-Raman spectra verify the phase-coexistence dependent water adsorption property in this material.

In the method section of manuscript:

Page 14: **Micro-Raman spectroscopy.** a Renishaw InVia confocal Raman spectrometer equipped with a Leica DMLM microscope was used to acquire Raman spectra. An Argon ion laser (wavelength 514.5 nm , model Stellar-REN, Modu-Laser) was used as an excitation source with an output power of 5 mW , and a 514.5 nm notch filter was adopted to remove the strong Rayleigh scattering. A 1800 g mm^{-1} (grooves per millimeter) diffraction grating yielded spectral resolution of $\sim 1\text{ cm}^{-1}$. The instrument was calibrated against Stokes Raman signal of pure Si at $520 \pm 0.05\text{ cm}^{-1}$ using a silicon wafer standard before performing measurements. The single crystals of lcp phase $1.9/2\text{H}_2\text{O}$ and nqp phase $1.4/3\text{H}_2\text{O}$ were located on the cover glass by tape with an optical microscope with a reflected light illumination attached to the Raman system. Laser beam was focused on the selected region of single crystal through a $50\times$ objective lens (0.75 numerical aperture), which provided spatial resolution of about $6\text{ }\mu\text{m}$. Spectroscopic measurements were made on samples with a diameter of about $40\text{ }\mu\text{m}$ chosen by the optical microscope. Spectra were collected for range $100\text{--}4000\text{ cm}^{-1}$ with three accumulations at 10 s acquisition time. Fixed single crystals were dispensed onto PTFE substrate in the sample cell at a different RH. Single crystal in phase-coexistence state was prepared by in situ drying the $1.9/2\text{H}_2\text{O}$ under a nitrogen flow.

In SI:

Supplementary Fig. 19 The reciprocal lattice and PXRD pattern of a phase-coexistence single crystal before and after by spraying water mist. A, Both the lattices of lcp and nqp phases can be reduced from the

diffraction data collected at 310 K. 1315 and 1632 diffraction peaks that belong to nqp and lcp phase, respectively, were indexed with 453 overlapped. The bottom shows the refined molecular frameworks of both lcp (left) and nqp (right) phases from the diffraction data. Only the lattice of lcp phase can be reduced from the diffraction data after spraying water mist on the single crystal for 30 min. The bottom shows the refined molecular frameworks of both lcp (left) and nqp (right) phases from the diffraction data. B, according to the orientations of crystal lattices in the phase-coexistence state deduced from reciprocal space analyses, the structures of nqp and lcp phases can be exactly fused together in a layer structure. The lattices and related peaks that belong to lcp and nqp phase are labelled in blue and red, respectively.

Supplementary Fig. 20 Micro-Raman spectra of 4-amino-1,2,4-triazole, cinamyle aldehyde, prentz ligand, $\text{K}_2\text{Pd}(\text{CN})_4$, lcp phase and nqp phase. The double peak at 1160 and 1178 cm^{-1} characterizes the structure of nqp phase and the broad peak at 1159 cm^{-1} characterizes the structure of lcp phase.

Supplementary Fig. 21 Micro-Raman spectra of pure nqp phase of 1-4/3H₂O. The peaks at 1160 and 1178 cm⁻¹ that denote to nqp structure persisted up to 3 days in saturated steam, and then converted to a single broad peak.

Supplementary Fig. 22 Experimental setup for Micro-Raman spectroscopy with different the different ambience.

Supplementary Fig. 23 Micro-Raman spectra of the phase-coexistence single crystal in different regions with different ambience. The Raman spectrum in panel I represents lcp phase with a small portion of nqp phase. The nqp phase in panels I–L changed to lcp within 10 min in a humidity of 80%.

Supplementary Fig. 17 PXRD patterns of pure nqp phase $1.4/3\text{H}_2\text{O}$ in saturated steam (A) and liquid water (B). The lcp phase $1.9/2\text{H}_2\text{O}$ can be recovered by leaving the pure nqp phase $1.4/3\text{H}_2\text{O}$ phase in saturated steam for ca. four days, while the phase-coexistence sample changes to pure lcp phase $1.9/2\text{H}_2\text{O}$ within 1 minute, suggesting the phase-coexistence state accelerates the water adsorption associated with nqp-to-lcp gate-opening transition. The all data were collected at room temperature after different activated treatment.

9. Page 7, lines 148-150: “the phase-coexistence state works on individual crystals rather than a physical mixture of two pure phases because the PXRD patterns of a physical mixture of the lcp phase $1.9/2\text{H}_2\text{O}$ and nqp phase $1.4/3\text{H}_2\text{O}$ did not show any significant change after spraying the water mist Supplementary Fig. 14)”. This conclusion is in contradiction to the water adsorption isotherm (Figure 2B) which clearly shows that nqp-to-lcp transition appears above $P/P_0 = 0.8$. Lack of significant changes in the PXRD may simply result from limited vapor diffusion or too short stabilization time.

Response: The sentence of Page 7, lines 148-150 ~~~ is not in contradiction to the water adsorption isotherm in Figure 2B, because the sample of Figure 2B, which was prepared by left the pure nqp phase in the steam atmosphere ($P/P_0 = 0.9$) for four days and then activated under vacuum at 298 K for 6 h, is in a phase-coexistence state. Although the PXRD pattern of physical-mixture sample is similar with the phase-

coexistence sample, the adsorption property of former is obviously different from the latter. Such distinct phenomenon is highly consistent with our assumption that the phase-coexistence state of individual crystals plays essential roles in the gate-opening adsorption.

To confirm the lack of significant changes in the PXRD pattern of physical-mixture sample is not own to

the limited vapor diffusion or too short stabilization time, we measured the water adsorption isotherm of this sample with the same experimental condition (activated under vacuum at 298 K for 6 h). As shown in Figure below (down panel of Figure 2A), the uptake of ~ 2.7 H_2O per Fe^{II} of this physical mixture sample is lower than that of ~ 2.7 H_2O per Fe^{II} for phase-coexistence sample (middle panel of Figure 2A), suggesting the nqp phase of physical-mixture sample cannot recover to lcp phase and phase-coexistence state of individual crystals plays essential roles in the gate-opening adsorption.

However, the difference between samples in Figure 2a and 2b is not well illuminated in our previous manuscript, therefore, some important information was added in the Figure 2 to avoid the confusion. Moreover, we think the different adsorption property of physical-mixture sample and phase-coexistence sample is important to support our assumption and we inserted the Supplementary Fig. 14 into Figure 2 of the manuscript. The changes are show below:

Fig. 2 Water adsorption isotherms and corresponding structural transformations probed by PXRD. **A**,

Isothermal vapor adsorption and desorption of the sample activated at 433 K under vacuum for 48 h (pure nqp phase, up), the sample activated at 298 K under vacuum (phase-coexistence state, middle), and a physical mixture sample of 1.9/2H₂O and 1.4/3H₂O activated at 298 K under vacuum (down). The solid and open curves denote the vapor adsorption and desorption, respectively. **B**, The continuous water adsorption-desorption cycles in different activated conditions. The abnormal drop of "Uptake" line marked by star at 1200 min is ascribed to the change of heating source (Furnace and water bath). **C**,

PXRD patterns upon water adsorption and desorption. 1.9/2H₂O loses water rapidly upon heating (323 K), which leads to a phase-coexistence state that can return to the lcp phase immediately by spraying with a water mist. Such rapid water adsorption of the phase-coexistence sample was performed two times to confirm the phenomenon. However, the completely dehydrated sample 1 prepared from long-time heating (433 K) under vacuum cannot recover to lcp phase smoothly upon water adsorption. In the measured sample, it took five days after leaving the pure 1 in saturated steam. The PXRD patterns of the physical mixture of two pure phases did not show any significant change after spraying the water mist for 60 min. The blue and red peaks denote the character peaks of structures of lcp and nqp phases, respectively.

11. Magnetic measurements seem to contradict the hypothesis of phase-coexistence in single crystals of 1. Figures 3B-D look like linear combinations of curves presented in Figure 3A and Figure 3E/3F. This would be highly unexpected for the phase-coexistence within single crystal, as SCO in this system is highly cooperative. Moreover, small changes in water content may very strongly influence SCO behavior, even in the absence of phase-coexistence phenomenon (well depicted by Song et al. in *Dalton Trans.* **45**, 18643-18652 (2016).).

Response: Indeed, small changes in water content may very strongly influence SCO behaviors, as shown in the papers by Song et al and ours (See ref. 60 and 61). In this material, the two-step SCO with a large plateau of lcp phase 1.9/2H₂O (**A**) was converted to a two-step SCO with no plateau of nqp phase 1.4/3H₂O upon water desorption, this phenomenon is consistent with previous reports.

As you pointed out, Figures 3B-D look like linear combinations of curves presented in Figure 3A and Figure 3E/3F, we think this abnormally phenomenon exactly reflects the phase-coexistence of individual crystals in this material. Based on our analyses, the simultaneous pore open and close accompanied with guest water redistribution leads to vary high energy barrier of 158 kcal mol⁻¹ that restrains the straightforward nqp phase to the lcp phase transformation. Hence, the phase-coexistence state with interfaces between the lcp and nqp phases is necessary to reduced the energy barrier through interfacial effect. The important roles of phase-coexistence state in the water adsorption was discussed above. As the diffusion of lattice from lcp to nqp phase was interrupted by the interfaces, both the structures of nqp and lcp phases were reserved in the crystals. Accordingly, a linear combination of curves presented in Figure 3A and Figure 3E/3F was detected in the samples presented in Figures 3B-D. It is worth mentioning that we tried dozens of times to detect the interface of single crystal under a polarizing microscope, but the results are not satisfied, because the interfaces may exist between the 2D molecular layers and our single crystal is too thin to perform the measurement.

According to your comment, we revised the related discussion in the manuscript to clarify this point:

Page 8: The SCO performances of phase-coexistence samples can be regarded as linear combinations of SCO transitions of 1.9/2H₂O and 1.4/3H₂O, and this phenomenon is significantly different from typical SCO compounds whose SCO behaviors are usually globally influenced by small changes in water content^{60,61}, suggesting the existence of interfaces between the lcp and nqp phases that interrupt the

diffusion of lattice from lcp to nqp phase. Notably, the transition temperatures and thermal hysteresis were slightly different

from that of $1.9/2\text{H}_2\text{O}$ and $1.4/3\text{H}_2\text{O}$, implying that the interfaces between the lcp and nqp phases of an individual crystal influence the SCO properties of each other⁶².

The paper you mentioned has been cited in the manuscript as references 60 and 61:

60 Wei, R. M. et al. Water induced spin-crossover behaviour and magneto-structural correlation in octacyanotungstate(iv)-based iron(ii) complexes. *Dalton Trans.* **45**, 18643-18652 (2016).

61 Yang, J. H., Zhao, Y. X., Xue, J. P., Yao, Z. S., & Tao, J. Reverse Hofmann-type spin-crossover compound showing a multichannel controllable color change in an ambient environment. *Inorg. Chem.* **60**, 7337-7344 (2021).

12. Sample preparation for magnetic measurements should be described in details in the discussion of magnetic properties (exact conditions of sample stabilization that lead in preparation of each phase mixture). This comment is answered together with comment 14.

14. Methods section should contain detailed description of sample packing for magnetic measurements. How was the sample stability achieved?

Response: The samples used for the magnetic measurements of Figures 4a and 4f were prepared from the standard method described above, while those for the magnetic measurement of Figures 4b-4e were prepared by heating the sample of $1.9/2\text{H}_2\text{O}$ at 323 K for different times (~ 2 min for 4b, ~4 min for 4c, ~ 6 min for 4d, ~ 8 min for 4e). The prepared samples were tightly wrapped with a plastic film ($2 \times 2 \text{ cm}^2$) and fixed in a straw. After that, the sample was loaded in the SQUID chamber at 250 K. The PXRD measurements were performed on the same samples to examine the ratios of nqp and lcp phases. In fact, both $1.9/2\text{H}_2\text{O}$ and $1.4/3\text{H}_2\text{O}$ are stable in the ambient condition (298 K, 60% humidity) according to TG and water adsorption measurements.

According to your suggestion, detailed informs about the sample preparation for magnetic measurements was added in both the main body of manuscript and method section as below:

In the manuscript:

Page 8: A multi-step SCO that combines the spin transitions of $1.9/2\text{H}_2\text{O}$ and $1.4/3\text{H}_2\text{O}$ was detected in the phase-coexistence samples. In order to ensure that the SCO behavior was consistent with the corresponding phase state, a certain amount of samples characterized by PXRD were taken and immediately folded and wrapped with plastic wrap for several times before compaction, and then were loaded in the magnetometer at the cabin temperature of 250 K.

In the method section:

Page 14: Magnetic measurements lcp phase $1.9/2\text{H}_2\text{O}$ and phase-coexistence states were performed on Quantum Design MPMS XL-7 magnetometer working in the 2–400 K temperature range with 2 K min^{-1} sweeping rate under a magnetic field of 5000 Oe. The samples used for the magnetic measurements of Figures 4a and 4f were prepared from the standard method described above, while those for the magnetic measurement of Figures 4b-4e were prepared by heating the sample of $1.9/2\text{H}_2\text{O}$ at 323 K for different times (~ 2 min for 4b, ~4 min for 4c, ~ 6 min for 4d, ~ 8 min for 4e). The prepared samples were tightly wrapped with a plastic film ($2 \times 2 \text{ cm}^2$) and fixed in a straw. After that, the sample was loaded in the SQUID chamber at 250 K. The PXRD measurements were performed on the same samples to examine the ratios of nqp and

lcp phases.

13. Methods section should contain description of the measurement setup (for HTK 1200N and TTK450) as well as sample preparation for PXRD measurements. Type of the sample holder and sample thickness are critical for assessing efficiency of vapor diffusion in such experiments.

Response: The detailed information about the measurement setup was added in the Methods section as below:

Page 12: **Powder X-ray diffraction (PXRD).** The variable-and room-temperature PXRD data were recorded on a PANalytical diffractometer with Cu K α radiation equipped with different temperature control parts: 1, the Anton Paar HTK 1200 accessory was designed for PXRD in reflection and transmission geometries with environmental heating of the sample up to 1473 K in air. The thermocouple was placed right underneath the round sample table (diameter ~1.5 cm, thickness ~0.5 mm and aluminium oxide material). 2, the Anton Paar TTK 450 chamber enabled in situ PXRD measurements in both reflection and transmission with the environmental temperature range from 100 K to 450 K under vacuum (liquid nitrogen cooling). The thermocouple was placed underneath the 1.4 x 1.8 cm² sample table (thickness ~1 mm and stainless steel material). For PXRD measurements that did not require temperature control part, the sample was loaded on the glass sample table with circular groove (diameter ~2 cm, thickness ~1 mm).

There are also several minor issues that should be corrected:

1. Page 2, line 16: "...leads to a pore rearrangement associated with local negative adsorption". No negative adsorption is observed for the presented network and therefore this part of the sentence is an unnecessary exaggeration. The observed behavior is pore rearrangement only and the new term "local negative adsorption" should not be coined.

Response: According to your suggestion, the "local negative adsorption" was deleted and the sentence have been modified as below:

Page 1: In this material, the adsorption-induced, non-uniform pedal motion of the axial prentz ligands and the crumpling/unfolding of the layer structure actuate a reversible narrow quasi-discrete pore (nqp) to large channel-type pore (lcp) change that leads to a pore rearrangement associated with simultaneous pore open and close.

Page 3: The "local negative adsorption" in the manuscript was also modified.

2. The first paragraph of the introduction section is not clear and especially lines 30-40 need rephrasing.

Response: The introduction section was rewritten as below:

Page 2: The development of flexible metal-organic frameworks (MOFs) or soft porous crystals (SPCs) whose porous structure can be reversibly altered in response to the sorption of guest molecules have attracted burgeoning attentions¹⁻⁵. Such adaptive structural transformability, usually ascribed to the

interplay between the guest molecules and flexible host frameworks, is associated with advanced properties in the field of gas separation, sensing, and magnetic switching⁶⁻¹⁰. In the past decades, many porous crystals that demonstrate guest-controllable porous expansion/contraction have been reported¹¹⁻²⁶. Two remarkable compounds are [Cu(aip)(H₂O)](solvent)_n and [Al(OH)_x(solvent)_y(TzDB)_z, (aip = 5-azidoisophthalate, TzDB = 4,4'-(1,2,4,5-tetrazine-3,6-diyl)dibenzoate), where the porous structure can be transformed by chemical reactions between host framework and guests, therefore exhibit intriguing self-accelerating absorption and domino-type porous transformation, respectively^{25,26}. These results encourage the further exploration of superior flexible MOFs whose porous structure can be reformed by unusual host-guest interactions. In

general, MOFs composed of conformation-variable ligands are susceptible to the chemical environmental variations induced by guest adsorption^{8,17,27-29}, particularly when the local rotation or reorientation of ligands can be propagated in the global structure through cooperative motions of the framework.

3. Page 3, line 31: “extend-retract motion¹²⁻¹⁴”. References 12-14 are not a good demonstration of this term.

Response: The introduction section was rewritten and the references 12-14 are as a part of references 11-26 about porous crystal that the guest-controllable porous expansion/contraction.

Lines 5-6, page 2: In the past decades, many porous crystals that demonstrate guest-controllable porous expansion/contraction have been reported¹¹⁻²⁶.

4. Page 4, line 75: “3D porous structure with channel pores (15.027 × 6.554 Å²)”. Even assuming that atoms are dimensionless, the pore dimensions would be 14.3 Å (for Pd-Pd) × 5.3 Å (for H-H of neighboring organic ligands). The authors should clearly specify which distances were listed as “channel pores” dimensions.

Response: We inserted a false atom in the center of the pore (Pd-Pd) by diamond software and determined the size of the pore by adjusting the diameter of the false atom (Fig. R2). The result was added as Supplementary Fig. 4 and the pore structure and void space of 1·9/2H₂O illustrated by the Connolly surface in yellow (probe of diameter 1.0 Å).

The papers about Pore structure and void space have been cited in the manuscript as references 47 and 48:

47 Zhang, Q., Chen, J., Zhu, X. C., Li, J., & Wu, D. 7-Connected Fe^{III}-based bio-MOF: Pore space partition and gas separations. *Inorg. Chem.* **59**, 16829-16832 (2020).

48 Qazvini, O. T., Babarao, R., Shi, Z. L., Zhang, Y. B., & Telfer, S. G. A robust ethane-trapping metal-organic framework with a high capacity for ethylene purification. *J. Am. Chem. Soc.* **141**, 5014-5020 (2019).

In the manuscript:

Page 3: The 2D coordination networks were assembled further via intermolecular interactions into a 3D porous structure with channel pores (13.2 × 4.4 Å²) extending along the crystallographic a-axis (Fig. 1C and Supplementary Fig. 4, regardless of the vander Waals radii)^{48,49}.

Fig. R2 The method of determining pore size.

Supplementary Fig. 4 Pore structure and void space of $1\cdot9/2\text{H}_2\text{O}$ illustrated by Connolly surface using a probe of diameter of 1 Å along a-axis.

5. On numerous occasions distances in the crystal structure are mentioned without uncertainty of their determination, which should be corrected.

Response: According to your suggestion, the distances in the crystal structure are remeasured with uncertainty of their determination and the sentences and figures have been modified.

Supplementary Fig. 7 Crystal structures and pore shapes of lcp phase $1\cdot9/2\text{H}_2\text{O}$ and nqp phase $1\cdot4/3\text{H}_2\text{O}$. (A) 3D framework of $1\cdot9/2\text{H}_2\text{O}$ with an interlayer distance of $14.618(3)$ Å. (B) 3D framework of $1\cdot4/3\text{H}_2\text{O}$ with an interlayer distance of $13.296(5)$ Å. (C) The large channel-type pores of $1\cdot9/2\text{H}_2\text{O}$ extending along the crystallographic a-axis. (D) The narrow quasi-discrete pore of $1\cdot4/3\text{H}_2\text{O}$ along the a-axis.

6. CHECKCIF report for 1-nqp at 100 K contains the following alert: PLAT920_ALERT_1_B Theta (Max)

in CIF and FCF Differ by 2.68 Degree. Please clarify what is the reason for that.

Response: The alert in the CIF file has been removed when the data reduction of 1-nqp at 100 K was re-performed using the HKL3000 software. This new result has been updated in CCDC.

7. Kepert *et al.* reported Hofmann-type frameworks based on very similar ligands and showing almost identical topology as 1-nqp (*Chem. Sci.* **9**, 5623-5629 (2018) and *Dalton Trans.* **50**, 1434-1442 (2021).) – those references should be mentioned in the introduction.

Response: The literatures you recommend have been cited in the introduction of the revised manuscript as new Ref. 44-46:

44 Ahmed, M. *et al.* Dual-supramolecular contacts induce extreme Hofmann framework distortion and multi-stepped spin-crossover. *Dalton Trans.* **50**, 1434-1442 (2021).

45 Zenere, K. A. *et al.* Increasing spin crossover cooperativity in 2D Hofmann-type materials with guest molecule removal. *Chem. Sci.* **9**, 5623-5629 (2018).

46 Brennan, A. T. *et al.* Guest removal and external pressure variation induce spin crossover in halogen-functionalized 2-D Hofmann frameworks. *Inorg. Chem.* **59**, 14296-14305 (2020).

8. I lack the expertise to evaluate details of DFT calculations, but I was alerted by the following description: “The experiment obtained two crystal structures [...] the other is lcp phase structure with 10 water

molecules in the unit cell, denoted as **lcp-10w**”. Why 10 water molecules were assumed in DFT calculations, whereas there are 9 water molecules in the crystal structure of 1-lcp?

Response: According to the SC-XRD data of lcp phase $1 \cdot 9/2\text{H}_2\text{O}$, some guest water molecules in the pores are disordered with a partial occupancy, which may affect the structural optimization implemented in the Vienna *ab initio* simulation package (VASP 5.4.4). According to the calculation result, lcp phase structure with 10 water molecules in the unit cell is reasonable for saturated adsorption state. More importantly, the number of water molecules, 9 or 10, has no influence on the procedure of adsorption-induced nqp-to-lcp lattice transformation.

This information was added in the SI as below:

The optimized lcp phase structure contains 10 water molecules in the unit cell, which reflects the disorder state of guest water molecules in the pores. The number of water molecules, 9 or 10, has no influence on the procedure of adsorption-induced nqp-to-lcp lattice transformation.

Response to reviewer #2:

Comments to Authors:

This manuscript reports a new example of an iron/palladium 2D Hoffmann network structure with 1D porosity. Such compounds have been heavily studied by the spin-crossover materials community for the past 20 years. They are well-known to afford interesting and useful thermal spin-transitions, which may be coupled to the removal or exchange of guest molecules within their pore structure. This is another example. It can be isolated in two different hydration states, which can be interconverted in single-crystal-to-single-crystal fashion. The two forms have very different pore structures and must interconvert via a detectable, metastable intermediate phase. That leads to gating during the interconversion of the two phases, which allows the mechanism of the interconversion to be monitored in unusual detail.

The design of the material is unexceptional, and single-crystal-to-single-crystal reactions in spin-crossover crystals are also quite well known now. Rather, the main interest lies in the complicated mechanism of the pore rearrangement between the phases, and how this manifests in the spin-transitions of the different phases. This has been thoroughly characterized by diffraction, by adsorption isotherms, and by periodic DFT calculations.

The authors have done a lot of experiments to sort this out. Some data aren't of the highest quality, but it's clear their interpretation and conclusions are sound. This is also novel enough for the journal, and it can be published if the following comments are addressed:

Response: *We thank this reviewer for positive comments on our work and valuable suggestions concerning the manuscript. Our point-by-point responses are provided below.*

1. The microanalyses of the two phases are good. However, a straightforward check would be to run thermogravimetric analyses (TGAs) on the two phases, as an extra confirmation of their water content. I was surprised those weren't included in the SI, and they should be added to the revision.

Response: *Thank you for pointing out the question. We performed the thermogravimetric (TG) measurements to further determine the guest content of the pristine crystal, lcp phase $1\cdot9/2\text{H}_2\text{O}$ and nqp phase $1\cdot4/3\text{H}_2\text{O}$.*

In the manuscript:

Page 3-4: *According to SC-XRD, elemental and thermogravimetric analyses, the pores accommodate $9/2$ H_2O molecules per Fe^{II} atom (Supplementary Fig. 3A).*

2nd paragraph, Page 4: *The narrow pores of this partially dehydrated crystal accommodate $4/3$ H_2O molecules per Fe^{II} atom according to the SC-XRD, elemental and thermogravimetric analyses (referred to $1\cdot4/3\text{H}_2\text{O}$, Supplementary Fig. 5B).*

In SI:

Supplementary Fig. 1 Thermogravimetric analysis of the pristine crystal. The pristine crystals prepared from MeOH/H₂O mixture contain both H₂O and MeOH molecules that can be partially exchanged when the sample is exposed in the air. The first step weight loss in the TG curve suggests a fast exchange and removal of MeOH guests in the room temperature, and the second step weight loss corresponds to the 1·9/2H₂O to 1·4/3H₂O transformation, and the third step weight loss indicates the 1·4/3H₂O to 1 transformation. Notably, although the TG measurement suggests the guest H₂O molecules were removed at ca. 400 K, a small part of 1·9/2H₂O (lcp phase structure) remains according to PXRD and vapor adsorption isotherm measurements.

Supplementary Fig. 4 Thermogravimetric analysis of lcp phase 1·4/3H₂O (A) and nqp phase 1·4/3H₂O (B). The anhydrous sample is stable up to ca. 503 K.

2. The crystal structures of the 1-nqp phase are of low quality. The 100 K structure is publishable, but the 250 K structure has very low data completeness, and too few observed data for a meaningful refinement

– the observed data : parameter ratio is only 6.5 : 1. That explains the very low precision of that structure (Table 5, supporting information). There is also obvious disorder in the prentz ligands at that temperature, which hasn't been modelled; in fact, it can't be modelled with such a weak dataset.

The basic interpretation of that structure is doubtless correct, but I wouldn't publish such a low quality crystal structure without a good reason. In fact, the 100 K structure gives all the information needed to

interpret the magnetic data of 1-nqp, so my choice would be to remove the 250 K structure from the manuscript.

In any case, the authors should comment on the low quality diffraction shown by 1-nqp – it's not truly a single-crystal-to-single-crystal transformation if the crystal degrades badly during the process.

Response: We remeasured the crystal structure of **1-nqp** ($1\cdot4/3\text{H}_2\text{O}$) at 250 K on a Rigaku Oxford XtaLAB PRO diffractometer equipped with graphite-monochromated Mo $K\alpha$ radiation ($\lambda = 0.71073 \text{ \AA}$). The quality of crystal structure is greatly improved with a data: $R_1 = 0.0984$ and data completeness = 99.7%, and the claim in the CIF file has been removed. These new results have been updated at CCDC and Tables 2, 5, and 6 in the revised supporting information.

In the manuscript:

2nd paragraph, page 8: The average Fe1–N bond lengths of **2.134(9) Å** at 250 K and **1.949(6) Å** at 100 K (Supplementary Table 5), respectively, suggest that Fe1 ion in $1\cdot4/3\text{H}_2\text{O}$ undergoes an complete SCO. While for Fe2 ion, an incomplete SCO is concluded as the average Fe2–N bond lengths shorten from **2.144(10) Å** at 250 K to **2.062(7) Å** at 100 K.

2nd paragraph, page 10: This is particular in the present case because the excluded water molecules exhibit strong hydrogen bonds with the N atoms of the framework with O...N distances of **2.754(2) Å** and **2.908(3) Å** (250 K, Fig. 4C and Supplementary Fig. 23 and Table 6).

In the SI:

Supplementary Table 2 Crystal data and structural refinements for $1\cdot4/3\text{H}_2\text{O}$.

	$1\cdot4/3\text{H}_2\text{O}$	
	100	250
Temperature/K	100	250
Formula	$\text{C}_{78}\text{H}_{68}\text{Fe}_3\text{N}_{36}\text{O}_4\text{Pd}_3$	
$M_r/\text{g mol}^{-1}$	2060.43	
Crystal size/mm	0.08x0.07x0.03	0.18x0.1x0.03
Wavelength/Å	0.77484	0.71073
Space group	$P\bar{1}$	$P\bar{1}$
Crystal color	red	yellow
Crystal system	triclinic	triclinic
a/Å	7.2100(14)	7.4258(5)
b/Å	16.097(3)	16.3023(11)
c/Å	19.051(4)	19.2325(12)
$\alpha/^\circ$	76.31(3)	76.768(6)
$\beta/^\circ$	80.61(3)	80.084(6)
$\gamma/^\circ$	79.15(3)	78.605(6)
Volume/Å ³	2093.5(8)	2202.0(3)
Z	1	1
$D_c/\text{g cm}^{-3}$	1.634	1.554
μ/mm^{-1}	1.210	1.150
F(000)	1036.0	1036.0
ω range/ $^\circ$	1.318 to 27.09	3.511 to 26.371
Data/restraints/parameters	7277/6/564	13208/18/500
Goodness-of-fit on F^2	1.129	1.032

Reflections collected	33449	13208
$R_1 [I \geq 2\sigma(I)]$	0.0881	0.0984
wR_2 [all data]	0.2671	0.2737
Largest diff. peak and hole/e.Å ⁻³	2.17/-2.33	4.94/-1.63
Completeness	89.0%	99.7%

Supplementary Table 5 Selected bond lengths and angles for 1·4/3H₂O.

	100K	250 K
Fe1–N1	1.940(5)	2.134(10)
Fe1–N2	1.941(6)	2.129(10)
Fe1–N7	1.965(6)	2.138(6)
Fe2–N3	2.059(5)	2.157(17)
Fe2–N4	2.039(7)	2.132(10)
Fe2–N5	2.052(8)	2.145(10)
Fe2–N6	2.061(7)	2.169(10)
Fe2–N11	2.096(8)	2.167(7)
Fe2–N15	2.062(6)	2.105(7)
C1–N1–Fe1	177.3(6)	173.2(11)
C2–N2–Fe1	177.5(6)	175.4(11)
C7–N7–Fe1	130.0(5)	128.1(5)
C3–N3–Fe2	175.3(6)	173.7(11)
C4–N4–Fe2	170.1(6)	168.7(11)
C5–N5–Fe2	176.5(7)	175.6(12)
C6–N6–Fe2	179.5(8)	177.7(11)
C18–N11–Fe2	123.1(6)	118.4(5)
C29–N15–Fe2	126.3(4)	126.2(5)

Supplementary Table 6 Selected structural parameters for 1·9/2H₂O and 1·4/3H₂O at different temperatures.

Compound	1·9/2H ₂ O				1·4/3H ₂ O	
	85 K	150 K	250 K	293 K	100 K	250 K
$\langle \text{Fe1–N} \rangle^{[a]}/\text{Å}$	1.963(4)	1.964(4)	2.167(5)	2.159(5)	1.949(7)	2.134(9)
$\langle \text{Fe2–N} \rangle^{[a]}/\text{Å}$	1.970(4)	2.160(4)	2.173(5)	2.169(5)	2.062(3)	2.144(9)
$\Sigma \text{Fe1}^{[b]}$	11.24(14)	8.28(18)	14.0(2)	12.92(18)	6.4(15)	13.2(6)
$\Sigma \text{Fe2}^{[b]}$	9.24(15)	14.04(17)	7.2(2)	5.12(18)	19.4(11)	11.5(4)
$d_{\text{N–O}}^{[c]}$	2.839(4)	2.847(4)	2.908(3)	2.896(4)	2.798(3)	2.754(2)
					2.940(4)	3.018(5)
$\langle \text{Fe1–N–C} \rangle^{[d]}$	179.20(5)	178.60(5)	178.41(3)	178.01(5)	177.4(5)	174.25(5)

$\langle \text{Fe2-N-C} \rangle^{[d]}$	172.56(2)	166.93(5)	167.26(3)	166.62(4)	175.35(3)	173.90(3)
$\Sigma \theta^{[e]}/^\circ$	4.72(6)	17.20(8)	7.46(5)	7.80(3)	5.043(12)	8.05(13)

[a] The average Fe–N bond lengths (Å);

[b] Octahedral distortion parameters ($^\circ$);

[c] The distance of hydrogen bonds between uncoordinated nitrogen atom of 1,2,4-triazole in the prentz ligand and oxygen atom of adjacent water;

[d] Average Fe–N–C angles within Hofmann layer;

[e] The parameter is defined as the average of the sum of $|90^\circ - \theta|$ for Pd–Fe–Pd and Fe–Pd–Fe angles within the Hofmann layer (Three Pd–Fe1–Pd, three Pd–Fe2–Pd, two Fe1–Pd–Fe1, two Fe2–Pd–Fe2 and two Fe2–Pd–Fe1 angles of 1·9/2H₂O; Three Pd–Fe1–Pd, five Pd–Fe2–Pd, two Fe1–Pd–Fe1, four Fe2–Pd–Fe2 and two Fe2–Pd–Fe1 angles of 1·4/3H₂O).

3. Why is the spin-transition at Fe(2) in 1-nqp incomplete? It probably isn't kinetically trapped, as the transition temperature is too high for that.

Response: We further measured the variable temperature magnetic susceptibilities of nqp phase 1·4/3H₂O and the $\chi_M T$ value showed 1.28 cm³ K mol⁻¹ at 50 K with different scan rates (10, 5, 2 and 1 K min⁻¹). The result suggested that the incomplete spin transition at Fe(2) was not kinetically trapped. The new data were added as Supplementary Fig. 28 and the main body of manuscript as below:

In the manuscript:

2nd paragraph, page 8: The $\chi_M T$ value shows unchange at 50 K with different scan rates (10, 5, 2 and 1 K min⁻¹), suggesting the incomplete spin transition of Fe2 was not kinetically trapped.

Supplementary Fig. 28 Temperature dependence of the $\chi_M T$ curves of nqp phase 1·4/3H₂O at different scan rates (10, 5, 2 and 1 K min⁻¹).

4. It looks like there is strong preferred orientation in the powder diffraction data in Figures 7-10 of the SI. That doesn't invalidate the measurements but it should be explained – were those samples true powders, or crystals?

Response: *The PXRD measurements were performed on finely grounded polycrystals. The strong preferred orientation in the powder diffraction data is due to the structural nature of 2D coordination network of the samples. The sentence was modified according to your suggestions.*

Page 6: When **the crystal sample of 1·9/2H₂O in lcp phase** was subjected to thermal treatment (> 323 K) or exposed to vacuum, peaks denoting the nqp phase appeared within 1 minute and increased gradually with time **(Fig. 2C and Supplementary Figs. 12–14).**

Changes made on the Supplementary Fig. 12 note:

Fig. 12 PXRD patterns of lcp phase 1·9/2H₂O upon heating under atmospheric condition. The peaks denoting the nqp phase appeared when the lcp phase was heated at 323 K for 1 min, and the phase-coexistence state persisted to 503 K. The PXRD data at each temperature were collected after a holding time of 10 min. **The PXRD measurements were performed on finely grounded polycrystals. The strong preferred orientation in the powder diffraction data is due to the structural nature of 2D coordination network of the samples.**

5. Figures 7-10 are also so compressed, that it's hard to see what's going on. I would expand them vertically, and just show one Figure per page, so we can see the powder patterns properly.

Response: *We have changed the layout of Figures 7-10 according to your suggestion in order to see the powder patterns properly.*

6. The text says Rietveld refinements were obtained for both phases (line 96), but only 1-lcp is included in the SI (Figure 18). The Rietveld refinement for 1-nqp should be added to the revision.

Response: *The Rietveld refinements for lcp phase 1·9/2H₂O and nqp phase 1·4/3H₂O at 250 K were show as Supplementary Fig. 9. Moreover, The Rietveld refinement for lcp phase 1·9/2H₂O at 100K was show as Supplementary Fig. 26 in order to confirm the purity of lcp phase 1·9/2H₂O during the SCO process.*

7. I know of one other spin-crossover material, where different phases coexist in the same crystal during a desolvation process (*Chem. Sci.* **7**, 2907 (2016)). That should be cited in the revision.

Response: *The literature you recommend has been cited in the revised manuscript as Ref. 62:*

62 *Aromí Bedmar, G. et al. Snapshots of a solid-state transformation: coexistence of three phases trapped in one crystal. Chem. Sci.* **7**, 2907-2915 (2016).

8. 'lcp' and 'nqp' are non-standard abbreviations. They are defined in the Abstract, but they should also be defined in the main text so the reader easily understands what they mean.

Response: *According to your suggestion, the abbreviations were added in the revised manuscript.*

Page 3: The atypical pore transformation is accompanied by a distinct adsorption process in which a stable phase-coexistence state, where narrow quasi-discrete pore (nqp) and large channel-type pore (lcp) phases are concomitant in an individual crystal, needs to be activated first by long-time exposure in saturated steam to accomplish the nqp-to-lcp gate-opening transition.

Reviewer #1 (Remarks to the Author):

The resubmitted manuscript contains improved experimental data when compared with the initial submission. Authors repeated single-crystal X-ray diffraction experiments using synchrotron radiation and conducted Raman spectroscopy experiment with dehydration/rehydration. They also included missing TGA experiments, additional water adsorption isotherm (Fig 2A, "Physical mixture") and clarification of some important experimental details. This allows for independent validation of experiments. Analysis of the data leads to the conclusion, that the described framework clearly demonstrates activation-method dependent adsorption properties. After long activation at 433 K in vacuum it shows only limited water adsorption, but water uptake is doubled in case of low temperature activation. This is depicted in Figure 2A and Figure 3.

I believe that experimental results presented in the revised manuscript are interesting, but I still disagree with the interpretation of these results. In my opinion additional measurements confirm lack of phase-coexistence within the single crystal. The observed sorption switching seems to result rather from the physical differences between anhydrous 1, $1\cdot4/3\text{H}_2\text{O}$ ("nqp phase") and $1\cdot9/2\text{H}_2\text{O}$ ("lcp phase"). This conclusion is based on the following points [enumeration of rows in the revised manuscript]:

1) Authors describe different methods of preparation for anhydrous 1 and for "nqp phase" which was characterized using scXRD as $1\cdot4/3\text{H}_2\text{O}$. The first one is prepared "by heating $1\cdot9/2\text{H}_2\text{O}$ at 433 K for 48 h under vacuum" and is only weakly crystalline ("an attempt to obtain the crystal structure of completely dehydrated crystal of 1 failed because of the poor diffraction") [lines 97-99]. Weak crystallinity of 1 is also reflected by large peak broadening in powder X-ray diffraction (Supplementary Fig. 8). On the other hand "crystals of the nqp phase $1\cdot4/3\text{H}_2\text{O}$ were obtained by heating $1\cdot9/2\text{H}_2\text{O}$ to 433 K for 48 h under vacuum and cooling in air." [lines 83-84]. This $1\cdot4/3\text{H}_2\text{O}$ "nqp phase" shows good crystallinity, as evidenced by single-crystal X-ray diffraction and PXRD pattern (Supplementary Fig. 9b). Therefore it is evident that 1 and $1\cdot4/3\text{H}_2\text{O}$ are not chemically identical. This is a very important observation for the later part of the discussion.

2) Authors claim that 1 and $1\cdot4/3\text{H}_2\text{O}$ share the same framework: "in situ generated 1 [...] possessed the same framework as that of $1\cdot4/3\text{H}_2\text{O}$ in the nqp phase (Supplementary Fig. 8)" [lines 96-98]. It is worth noting, that they are not identical – experimental PXRD pattern for 1 depicted in Supplementary Fig. 8 differs from the simulated pattern for $1\cdot4/3\text{H}_2\text{O}$ and these differences do not seem to be caused by preferred orientation of crystallites only. It may be difficult to evaluate due to the peak broadening observed for 1. Still, I believe that 1 and $1\cdot4/3\text{H}_2\text{O}$ share similar "narrow quasi-discrete pore" structure type, but they are not exactly the same phases.

3) 1 and $1\cdot4/3\text{H}_2\text{O}$ share the same position of the first diffraction peak at around 6.6 degrees 2θ . Therefore it is impossible to distinguish 1 from $1\cdot4/3\text{H}_2\text{O}$ by comparing 5-15 degrees 2θ range only, despite of the fact that they are two chemically different phases (as explained in point 1).

4) Figure 2A, top panel: The water sorption isotherm is labelled as "Pure nqp phase". Throughout the manuscript authors use "nqp phase" for the description of $1\cdot4/3\text{H}_2\text{O}$. On the other hand, caption to Figure 2A reads: "A, Isothermal vapor adsorption (solid red) and desorption (open red) of the sample activated at 433 K under vacuum for 48 h (up)". This is supported by: "In the first measurement, the $1\cdot9/2\text{H}_2\text{O}$ sample was activated in situ at 433 K under vacuum for 48 h and then measured at 298 K" [lines 311-312] These activation conditions do not lead to $1\cdot4/3\text{H}_2\text{O}$, but to anhydrous 1, as explained in point 1 of this review and specified by authors themselves (lines 97-99). Therefore I believe that top panel of Figure 2A shows pore filling of "nqp structure type", which starts with compound 1 at $p/p_0 = 0$ and proceeds to $\approx 4/3\text{H}_2\text{O}$ molecules per Fe water uptake at $p/p_0 = 0.9$, which corresponds to compound $1\cdot4/3\text{H}_2\text{O}$. Moreover, please note that water adsorption isotherm starts with a flat plateau in $p/p_0 = 0-0.08$ range. This adsorption behavior may indicate transition from closed pore structure to the narrow quasi-discrete pore structure, corresponding to the transition from 1 to $1\cdot4/3\text{H}_2\text{O}$. As there is no such plateau in the water desorption isotherm, structure would probably remain in this narrow

quasi-discrete pore structure type in the desorption branch. This is confirmed by powder X-ray diffraction results. PXRD pattern of the sample after isothermal cycle of water sorption for the sample activated at 433 K under vacuum for 48 h (Supplementary Fig. 11) is not the same as PXRD pattern for the sample activated at 433 K under vacuum for 48 h (Supplementary Fig. S8).

5) Figure 2A, middle panel: The water sorption isotherm was measured for “the 1·9/2H₂O sample activated at 298 K under vacuum for 6 h” [lines 314-315]. State of the sample after this kind of activation cannot be determined reliably, as there is no PXRD pattern registered under exactly the same conditions and thus my conclusions are based on water sorption isotherm. Please note, that there is no sign of inflection at $p/p_0 = 0.1$. While authors attribute this to the appearance of the phase-coexistence state of 1·4/3H₂O and 1·9/2H₂O, I believe this results simply from the absence of anhydrous 1 in that sample. The shape of the water isotherm is well explained by initial monolayer adsorption within the evacuated “large channel pore” structure type at low p/p_0 , which is followed by hysteretic capillary condensation in the micropores.

6) Figure 2A, bottom panel: “isothermal vapor adsorption (solid blue) and desorption (open blue) of the physical mixture (1·9/2H₂O and 1·4/3H₂O) at 298 K under vacuum (down)” [lines 553-554; the caption should read “green”, not “blue”]. In fact, this looks like combination of water isotherm from top panel (which is 1, NOT 1·4/3H₂O – please see point 4 of this review) and middle panel. To summarize, this is the expected phase identity of the sample at the beginning of each measurement ($p/p_0 = 0$): top panel – anhydrous 1 (in line with activation at 433 K under vacuum for 48 h), middle panel – evacuated 1·9/2H₂O (probably still possessing lcp structure, in line with the natural shape of water isotherm for a microporous material), bottom panel – mixture of anhydrous 1 (characterized by the inflection at $p/p_0 = 0.1$) and evacuated 1·9/2H₂O. “Pure nqp phase” (1·4/3H₂O) is present only at $p/p_0 = 0.9$ on the top panel.

Remarks 1-6 lead me to the following conclusion: 1·9/2H₂O shows the “large channel pore” type of structure, where approximately 3 water molecules per Fe can be evacuated in mild conditions with little effect to the coordination skeleton. When this compound is heated at 433 K for a long time it is transformed into anhydrous 1, which was not characterized by scXRD, but possibly has the “narrow quasi-discrete pore” structure, or even shows completely closed pores (please see point 4 of this review). Anhydrous 1 relatively quickly absorbs approximately 1.5 water molecule per Fe, at high humidity producing well-crystalline 1·4/3H₂O. However, transformation of “nqp” structure to “lcp” structure requires several days in high humidity, producing fully hydrated 1·9/2H₂O. This fully explains top panel of Figure 2A (which shows equilibrium between anhydrous 1 and structurally-characterized 1·4/3H₂O) and middle panel of Figure 2A (which shows equilibrium between evacuated “lcp” structure and structurally-characterized 1·9/2H₂O). Slow dynamics between nqp and lcp structures (which is in line with high nqp-lcp energy barrier calculated by the authors) fully explains water sorption data, without a need to employ the elusive “phase-coexistence state”. Moreover, other measurements also indicate lack of this phase-coexistence within a single crystal:

7) In accordance with my previous review, I believe that phase-coexistence within a single crystal can be tested only with a spatially resolved method. Authors utilized Raman spectroscopy to conduct such an experiment on a single crystal. Again the identity of the “nqp phase” is not clear. It was dried at 433 K for 48 hours (Supplementary Fig. 21), which depending on the cooling conditions (dry or humid air) would produce anhydrous 1 or 1·4/3H₂O, but cooling conditions were not specified. Regardless of the hydration state, both these compounds are expected to show “nqp” structure, which very slowly transforms into “lcp” structure in water vapor. This explains why 4 days in saturated stream were required to observe the transformation.

8) Figure 3 shows spatially resolved experiment, where 4 different areas of a single crystal were tested (C-F). All these positions shows very similar Raman spectra after N₂ purge (G-J) and exactly identical Raman spectra after rehydration (K-N). This is exactly opposite, to what should be observed in case of phase-coexistence. In such a case, spectra depicted in Fig. 3G-J would be very different, as some of them would resemble “lcp” structure type, and the others would resemble “nqp” structure type. In fact, no spectra in Fig. 3 resembles “nqp” structure type. All spectra in the Fig. 3G-N show the same intensity of two bands around 1400 cm⁻¹. For comparison, in the Raman spectrum of “nqp” structure depicted in Supplementary Fig. 21 band at ≈1390 cm⁻¹ has twice the intensity of that at

$\approx 1410 \text{ cm}^{-1}$. The variation between Figure 3G-J and K-N seems to result from differences between fully hydrated “lcp structure” ($1.9/2\text{H}_2\text{O}$) and evacuated “lcp” structure, with little effect to the coordination skeleton and no presence of $1.4/3\text{H}_2\text{O}$. Again, this experiment demonstrated different water adsorption dynamics for “nqp” structure and “lcp” structure, but not the existence of “nqp” and “lcp” structure within a single crystal.

9) One may argue that PXRD patterns combined with adsorption measurements seem to confirm phase-coexistence conclusion. However, as noted in point 3 of this review, the first diffraction peak is very misleading, as it is in the same position for 1 and $1.4/3\text{H}_2\text{O}$. Please see for example Supplementary Fig. S15 – only two phases are accounted for: $1.9/2\text{H}_2\text{O}$ “blue” and $1.4/3\text{H}_2\text{O}$ “red”. Thus, the same “red” phase ($1.4/3\text{H}_2\text{O}$) is assigned to patterns labelled as 360 min. and 2880 min., despite the vast difference in 15-30 degrees 2θ region. This makes analysis of this first diffraction peak in PXRD patterns very misleading and it may result in wrong conclusions about phase identity of the sample. I believe that PXRD pattern labelled as 2880 min. was obtained for the sample mostly composed of anhydrous 1.

10) “The SCO performances of phase-coexistence samples can be regarded as linear combinations of SCO transitions of $1.9/2\text{H}_2\text{O}$ and $1.4/3\text{H}_2\text{O}$, and this phenomenon is significantly different from typical SCO compounds whose SCO behaviors are usually globally influenced by small changes in water content, suggesting the existence of interfaces between the lcp and nqp phases that interrupt the diffusion of lattice from lcp to nqp phase.” [lines 214-221]

I strongly disagree with this conclusion. When properties of the mixture resemble linear combination of the ingredients, this means that there is no interaction between the ingredients. Physical mixture of $1.9/2\text{H}_2\text{O}$ and FeCl_2 will show linear combination of their magnetic properties, despite of the fact that $1.9/2\text{H}_2\text{O}$ and FeCl_2 do not show phase-coexistence within single crystals. It is well known that cooperative SCO (as observed for $1.9/2\text{H}_2\text{O}$) is strongly affected by variation in crystallite size and similar interface effects. Thus existence of the interface between the lcp and the nqp is expected to affect magnetic properties of both phases, which is not observed in the data.

In conclusion, I believe that the resubmitted manuscript contains mostly valid measurements (unfortunately PXRD patterns suffer from strong texture, but that results from the experimental setup) and the observed phenomenon of “programmable” sorption is interesting, with the “lcp” structure showing twice the water uptake of the “nqp” structure. However, analysis of the resubmitted version led me to the conclusion, that this behavior can be fully explained by slow lcp-nqp dynamics. In my opinion there is no evidence for phase-coexistence within the single crystals of this material. Therefore I encourage the authors to re-analyze relevant measurements and prepare a revised text, with emphasis on the interconversion between anhydrous 1, “nqp” structure type and “lcp” structure type”, rather than questionable phase-coexistence phenomenon (in my opinion no additional measurements are necessary to fully describe behavior of this framework in the absence of phase-coexistence hypothesis).

Additional comments:

11) “the decomposition of framework of 1 should be precluded, as we can re-obtain the crystal structure of $1.4/3\text{H}_2\text{O}$ by exposing one of the above single crystals in the air” [rebuttal, page 11]

The ability to re-obtain the crystalline form by re-solvation does not confirm the stability of the parent framework. Recently there were several reports regarding solvent-driven transformation from amorphous to crystalline state (Dalton Trans., 2018, 47, 845; J. Am. Chem. Soc. 2021, 143, 20202 and for cyanides: Chem. Sci., 2021, 12, 9176).

12) Supporting Fig. 11 – it would be more appropriate to compare the sample after sorption experiments with the already activated sample (Supp. Fig. S11A – I believe that PXRD pattern from Supp. Fig. S8 may be used for that).

13) On several occasions experimental patterns were compared to the simulated pattern for 1.4/3H₂O at 100 K (for example Supp. Fig. S13) – please replace it with the simulated pattern for structure at 250 K.

14) Fig. 2B – “activation” arrow should end around 1300 minutes.

15) Article, page 14 – in the description of Raman setup millimeters should probably be replaced with micrometers.

Reviewer #2 (Remarks to the Author):

Many of reviewer 1’s comments on the original manuscript, and some of mine, concerned the crystallography. That has been addressed by remeasuring or re-refining all the structures in the study, including some new data collections with synchrotron radiation. The new structural data are a significant improvement on the originals, and the many of the experimental issues itemised by reviewer 1 have now been addressed. The new crystal structures from the nqp phase are of reasonable quality, for the product of a single-crystal-to-single-crystal annealing reaction.

The revision also includes new thermogravimetric analyses and extra magnetic data (which I requested), and single crystal Raman microscopy (to address a comment by reviewer 1). Those are all helpful additions.

I have just a few new comments to address before final acceptance of the manuscript.

(i) My only criticism of the new data, is that the Raman microscopy data in Supplementary Figure 23 don’t clearly show the co-existence of both phases in the same crystal, as proposed in the caption. Spectra E-H all look the same; I-L all look the same; and M-P all look the same. So, the phase composition at all four points of the crystal looks the same, under each of the conditions measured.

So, that experiment was worth doing, but it doesn’t prove the two phases can co-exist in the same crystal, at the same time. At best, what it shows is that the domains of the two phases in the mixed-phase crystal must be smaller than the resolution of the Raman microscope. The caption to Supplementary Figure 23, and the description on lines 167-175, should be changed to reflect that.

Supplementary Figure 19 does show the two phases can co-exist in the same crystal, though, so that conclusion is sound on a macroscopic level, even if it was not observed microscopically.

(ii) Line 153. The abbreviation ‘SCP’ is not in common use, and isn’t defined anywhere. The ‘CP’ is presumably Coordinated Polymer, but I don’t know what the ‘S’ stands for. Since it isn’t used anywhere else in the manuscript, that abbreviation should be changed for something more obvious.

(iii) An English error that comes up several times is “pore open and close”, which should be changed to “pore opening and closing”.

More generally, while the manuscript is always understandable, there are many small English mistakes and the text would benefit from English corrections at the proof stage.

Responses to reviewers' comments

We are grateful to all reviewers for their thorough evaluation of our revised manuscript "*Single crystal-to-single crystal and spin-crossover transformations via phase-coexistence state in pore-adjustable frameworks*" (NCOMMS-21-40232A). Their kind comments and constructive suggestions would highly help us improve the quality of our manuscript. Now, we have completed revision and addressed all questions raised by all reviewers. We have changed the manuscript title into "*A spin-crossover framework endowed with pore-adjustable behavior by slow structural dynamics*". In this 2nd revised manuscript and supplementary material, all changes have been highlighted in yellow. Our point-by-point responses to all reviewers' comments are listed below, shown in italic and red.

Responses to reviewer #1:

Comments to Authors:

The resubmitted manuscript contains improved experimental data when compared with the initial submission. Authors repeated single-crystal X-ray diffraction experiments using synchrotron radiation and conducted Raman spectroscopy experiment with dehydration/rehydration. They also included missing TGA experiments, additional water adsorption isotherm (Fig 2A, "Physical mixture") and clarification of some important experimental details. This allows for independent validation of experiments. Analysis of the data leads to the conclusion, that the described framework clearly demonstrates activation-method dependent adsorption properties. After long activation at 433 K in vacuum it shows only limited water adsorption, but water uptake is doubled in case of low temperature activation. This is depicted in Figure 2A and Figure 3.

I believe that experimental results presented in the revised manuscript are interesting, but I still disagree with the interpretation of these results. In my opinion additional measurements confirm lack of phase-coexistence within the single crystal. The observed sorption switching seems to result rather from the physical differences between anhydrous **1**, $1\cdot4/3\text{H}_2\text{O}$ ("nqp phase") and $1\cdot9/2\text{H}_2\text{O}$ ("lcp phase"). This conclusion is based on the following points:

Response: *Thanks for providing your meticulous and very valuable comments. We agree with your viewpoint of slow lcp-nqp dynamics, so we have carried out comprehensive consideration and made detailed changes about the "phase-coexistence state" description.*

We now use "partially dehydrated sample" instead of 'phase-coexistence state' to refer to the samples with mild activation, and emphasize "activation-method dependent adsorption properties are caused by slow nqp-lcp dynamics and essentially determined by the pore rearrangement associated with simultaneous pore opening and closing." in the revised manuscript.

The phrase "partially dehydrated sample" conforms to the core point of slow lcp-nqp dynamics, the reasons are shown below: 1) When the crystal sample of $1\cdot9/2\text{H}_2\text{O}$ in lcp phase was subjected to mild activation, i.e., thermal treatment (323 K) or exposed to vacuum, new peaks appeared in the higher-angle region within 1 minute and increased gradually with time, indicating the contraction of partially dehydrated framework (Fig. R1, ref. Science 2010, 329, 1053). 2) the lcp structural characteristics can persist in the sample prepared under mild activating conditions. 3) According to the TG of lcp phase $1\cdot9/2\text{H}_2\text{O}$, a small amount of solvent remains when the crystal sample of $1\cdot9/2\text{H}_2\text{O}$ in lcp phase was subjected to mild activation.

Fig. R1 Comparison of the PXR D pattern of lcp phase 1·9/2H₂O under vacuum with the simulated PXR D pattern of lcp phase 1·9/2H₂O and the simulated PXR D pattern of nqp phase 1·4/3H₂O.

1. Authors describe different methods of preparation for anhydrous **1** and for “nqp phase” which was characterized using SCXRD as 1·4/3H₂O. The first one is prepared “by heating 1·9/2H₂O at 433 K for 48 h under vacuum” and is only weakly crystalline (“an attempt to obtain the crystal structure of completely dehydrated crystal of **1** failed because of the poor diffraction”) [lines 97-99]. Weak crystallinity of **1** is also reflected by large peak broadening in powder X-ray diffraction (Supplementary Fig. 8). On the other hand “crystals of the nqp phase 1·4/3H₂O were obtained by heating 1·9/2H₂O to 433 K for 48 h under vacuum and cooling in air.” [lines 83-84]. This 1·4/3H₂O “nqp phase” shows good crystallinity, as evidenced by single-crystal X-ray diffraction and PXR D pattern (Supplementary Fig. 9b). Therefore it is evident that **1** and 1·4/3H₂O are not chemically identical. This is a very important observation for the later part of the discussion.

Response: We agree that **1** and 1·4/3H₂O are not chemically identical. The sentences “Powder X-ray diffraction (PXR D) experiments revealed that in situ generated **1** by heating 1·9/2H₂O at 433 K for 48 h under vacuum possessed the same framework as that of 1·4/3H₂O in the nqp phase (Supplementary Fig. 8).” have now been changed into “Although Powder X-ray diffraction (PXR D) experiments revealed that in situ generated **1** by heating 1·9/2H₂O at 433 K for 48 h under vacuum possessed the similar contracted structure as that of nqp phase 1·4/3H₂O, **1** and 1·4/3H₂O were not exactly the same phases (Supplementary Fig. 8).” in the revised manuscript (lines 16-18, Page 4).

2. Authors claim that **1** and 1·4/3H₂O share the same framework: “in situ generated **1** [...] possessed the same framework as that of 1·4/3H₂O in the nqp phase (Supplementary Fig. 8)” [lines 96-98]. It is worth noting, that they are not identical – experimental PXR D pattern for **1** depicted in Supplementary Fig. 8 differs from the simulated pattern for 1·4/3H₂O and these differences do not seem to be caused by preferred orientation of crystallites only. It may be difficult to evaluate due to the peak broadening observed for **1**. Still, I believe that **1** and 1·4/3H₂O share similar “narrow quasi-discrete pore” structure type, but they are not exactly the same phases.

Response: The caption for Supplementary Fig. 8 has been changed into “**Supplementary Fig. 8** In situ PXR D patterns of **1** upon heating the dried sample at 433 K for 48h under vacuum (Green, the heating rate

of 5 K min⁻¹) compared with that of 1·4/3H₂O (Red). The PXRD data of 1 was collected at room temperature after activated treatment.”.

3. 1 and 1·4/3H₂O share the same position of the first diffraction peak at around 6.6 degrees 2theta. Therefore it is impossible to distinguish 1 from 1·4/3H₂O by comparing 5-15 degrees 2theta range only, despite of the fact that they are two chemically different phases (as explained in point 1).

Response: Thank you for your comment. By comparing 5–40 degrees 2theta range of 1 and 1·4/3H₂O, they are not exactly the same phases (Supplementary Fig. 8).

4. Figure 2A, top panel: The water sorption isotherm is labelled as “Pure nqp phase”. Throughout the manuscript authors use “nqp phase” for the description of 1·4/3H₂O. On the other hand, caption to Figure 2A reads: “A, Isothermal vapor adsorption (solid red) and desorption (open red) of the sample activated at 433 K under vacuum for 48 h (up)”. This is supported by: “In the first measurement, the 1·9/2H₂O sample was activated in situ at 433 K under vacuum for 48 h and then measured at 298 K” [lines 311-312] These activation conditions do not lead to 1·4/3H₂O, but to anhydrous 1, as explained in point 1 of this review and specified by authors themselves (lines 97-99). Therefore I believe that top panel of Figure 2A shows pore filling of “nqp structure type”, which starts with compound 1 at p/p₀ = 0 and proceeds to ≈4/3H₂O molecules per Fe water uptake at p/p₀ = 0.9, which corresponds to compound 1·4/3H₂O. Moreover, please note that water adsorption isotherm starts with a flat plateau in p/p₀ = 0–0.08 range. This adsorption behavior may indicate transition from closed pore structure to the narrow quasi-discrete pore structure, corresponding to the transition from 1 to 1·4/3H₂O. As there is no such plateau in the water desorption isotherm, structure would probably remain in this narrow quasi-discrete pore structure type in the desorption branch. This is confirmed by powder X-ray diffraction results. PXRD pattern of the sample after isothermal cycle of water sorption for the sample activated at 433 K under vacuum for 48 h (Supplementary Fig. 11) is not the same as PXRD pattern for the sample activated at 433 K under vacuum for 48 h (Supplementary Fig. S8).

Response: Thank you for your meticulous examination. As you suggested, we have rearranged the structural transformation during the activation-method dependent adsorption. This information has been amended in the Figure 2 legend and Supplementary Fig. 18. The sentences “After that, it showed water adsorption with a threshold pressure of P/P₀ = 0.08 (Fig. 2A, up).” have been changed into “After that, the water adsorption isotherm started with a flat plateau in P/P₀ = 0–0.08 range. The adsorption behavior may indicate structural transformation from closed framework to nqp framework, corresponding to the transition from 1 to 1·4/3H₂O (Fig. 2A, up).” (lines 3-6, page 5).

In addition, the caption for Figure 2 has also been modified, see below marked in yellow.

Fig. 2 Water adsorption isotherms and corresponding structural transformations probed by PXRD. **A**, Isothermal vapor adsorption (solid red) and desorption (open red) of the sample activated at 433 K under vacuum for 48 h (up), isothermal vapor adsorption (solid blue) and desorption (open blue) of the sample activated at 298 K under vacuum (middle) and isothermal vapor adsorption (solid green) and desorption (open green) of the physical mixture (1·9/2H₂O and 1·4/3H₂O) at 298 K under vacuum (down). **B**, The continuous water adsorption-desorption cycles in different activated conditions. The abnormal drop of “Uptake” line marked by star at 1200 min is ascribed to the change of heating source (Furnace and water bath). **C**, PXRD patterns upon water adsorption and desorption. 1·9/2H₂O loses water rapidly upon heating (323 K), which leads to a partially dehydrated sample that can return to the lcp phase immediately by spraying with a water mist. Such rapid water adsorption of the partially dehydrated sample was performed two times to confirm the phenomenon. However, the completely dehydrated sample 1 prepared from long-time heating (433 K) under vacuum can rehydrate to nqp phase 1·4/3H₂O within 60 min, while cannot recover to lcp phase smoothly upon water adsorption. In the measured sample, it took four days after leaving the 1 or 1·4/3H₂O in saturated steam. The PXRD patterns of the physical mixture of two phases did not show any significant change after spraying the water mist for 60 min. The gray, blue and red peaks denote the

character peaks of structures of partially dehydrated, lcp and nqp framework, respectively.

Supplementary Fig. 18 has also been modified, as shown below:

Supplementary Fig. 18 Activation-method dependent water-adsorption (Red lines: desorption; Blue lines: adsorption).

5. Figure 2A, middle panel: The water sorption isotherm was measured for “the 1·9/2H₂O sample activated at 298 K under vacuum for 6 h” [lines 314-315]. State of the sample after this kind of activation cannot be determined reliably, as there is no PXRD pattern registered under exactly the same conditions and thus my conclusions are based on water sorption isotherm. Please note, that there is no sign of inflection at $p/p_0 = 0.1$. While authors attribute this to the appearance of the phase-coexistence state of 1·4/3H₂O and 1·9/2H₂O, I believe these results simply from the absence of anhydrous 1 in that sample. The shape of the water isotherm is well explained by initial monolayer adsorption within the evacuated “large channel pore” structure type at low p/p_0 , which is followed by hysteretic capillary condensation in the micropores.

Response: According to the TG measurement, the sample is partially dehydrated and we agree that the partially dehydrated sample does not contain the anhydrous 1. We think that “partially dehydrated sample/framework” is a better description of the sample with mild activation than “phase-coexistence state” or “evacuated lcp structure type”. Firstly, we agree that the sample with mild activation has the structural characteristics of lcp structure type, which is not contradictory with the “partially dehydrated sample/framework”. Secondly, a small amount of solvent remains in the samples with mild activation. Thirdly, when the crystal sample of 1·9/2H₂O in lcp phase was subjected to mild activation, i.e., thermal treatment (323 K) or exposed to vacuum, new peaks appeared in the higher-angle region within 1 minute and increased gradually with time, indicating the contraction of partially dehydrated framework (Fig. R1, ref. Science 2010, 329, 1053).

The sentences “When the crystal sample of 1·9/2H₂O in lcp phase was subjected to thermal treatment (> 323 K) or exposed to vacuum, peaks denoting the nqp phase appeared within 1 minute and increased gradually with time (Fig. 2C and Supplementary Figs. 12–14).” and “These results suggest that the lcp phase can persist in the sample prepared under mild activating conditions, and such a lcp and nqp phase-coexistence state plays a vital role in the guest adsorption associated with pore transformation from the nqp phase to the lcp phase³¹.” have been changed into “When the crystal sample of 1·9/2H₂O in lcp phase was subjected to mild activation, i.e., thermal treatment (323 K) or exposed to vacuum, new peaks appeared in the higher-angle region within 1 minute and increased gradually with time, indicating the contraction of partially dehydrated framework (Fig. 2C and Supplementary Figs. 5, 12–14).” [lines 5-8, Page 6].

These results suggest that the lcp structural characteristics of partially dehydrated framework can persist in the sample prepared under mild activating conditions, and such flexible framework plays a vital role in the doubled water adsorption⁵¹⁻⁵⁵. [lines 18-21, page 6]

6. Figure 2A, bottom panel: “isothermal vapor adsorption (solid blue) and desorption (open blue) of the physical mixture (1·9/2H₂O and 1·4/3H₂O) at 298 K under vacuum (down)” [lines 553-554; the caption should read “green”, not “blue”]. In fact, this looks like combination of water isotherm from top panel (which is 1, NOT 1·4/3H₂O – please see point 4 of this review) and middle panel. To summarize, this is the expected phase identity of the sample at the beginning of each measurement (p/p₀ = 0): top panel – anhydrous 1 (in line with activation at 433 K under vacuum for 48 h), middle panel – evacuated 1·9/2H₂O (probably still possessing lcp structure, in line with the natural shape of water isotherm for a microporous material), bottom panel – mixture of anhydrous 1 (characterized by the inflection at p/p₀ = 0.1) and evacuated 1·9/2H₂O. “Pure nqp phase” (1·4/3H₂O) is present only at p/p₀ = 0.9 on the top panel.

Remarks 1-6 lead me to the following conclusion: 1·9/2H₂O shows the “large channel pore” type of structure, where approximately 3 water molecules per Fe can be evacuated in mild conditions with little effect to the coordination skeleton. When this compound is heated at 433 K for a long time it is transformed into anhydrous 1, which was not characterized by scXRD, but possibly has the “narrow quasi-discrete pore” structure, or even shows completely closed pores (please see point 4 of this review). Anhydrous 1 relatively quickly absorbs approximately 1.5 water molecule per Fe, at high humidity producing well-crystalline 1·4/3H₂O. However, transformation of “nqp” structure to “lcp” structure requires several days in high humidity, producing fully hydrated 1·9/2H₂O. This fully explains top panel of Figure 2A (which shows equilibrium between anhydrous 1 and structurally-characterized 1·4/3H₂O) and middle panel of Figure 2A (which shows equilibrium between evacuated).

Response: *Thank you for your suggestion. We now use ‘partially dehydrated sample’ instead of ‘phase-coexistence state’ to refer to the samples with mild activation, which has the lcp structural characteristics.*

7. In accordance with my previous review, I believe that phase-coexistence within a single crystal can be tested only with a spatially resolved method. Authors utilized Raman spectroscopy to conduct such an experiment on a single crystal. Again the identity of the “nqp phase” is not clear. It was dried at 433 K for 48 hours (Supplementary Fig. 21), which depending on the cooling conditions (dry or humid air) would produce anhydrous 1 or 1·4/3H₂O, but cooling conditions were not specified. Regardless of the hydration state, both these compounds are expected to show “nqp” structure, which very slowly transforms into “lcp” structure in water vapor. This explains why 4 days in saturated stream were required to observe the transformation.

Response: *The nqp single crystal used for Raman measurement was obtained by heating 1·9/2H₂O to 433 K for 48 h under vacuum and cooling in air to rehydration. The sentences “The crystals of the nqp phase 1·4/3H₂O were obtained by heating 1·9/2H₂O to 433 K for 48 h under vacuum and cooling in air. Upon water desorption, the structural transformation can be fully characterized at the atomic level by SC-XRD owing to the single-crystal-to-single-crystal nature.” have been changed into “The crystals of the nqp phase 1·4/3H₂O were obtained by heating 1·9/2H₂O to 433 K for 48 h under vacuum and cooling in air to rehydration. Upon water desorption and re-adsorption, the structural transformation can be fully characterized at the atomic level by SC-XRD owing to the single-crystal-to-single-crystal nature.” [lines 3-5, page 4].*

8. Figure 3 shows spatially resolved experiment, where 4 different areas of a single crystal were tested (C-F). All these positions shows very similar Raman spectra after N₂ purge (G-J) and exactly identical Raman spectra after rehydration (K-N). This is exactly opposite, to what should be observed in case of phase-coexistence. In such a case, spectra depicted in Fig. 3G-J would be very different, as some of them would resemble “lcp” structure type, and the others would resemble “nqp” structure type. In fact,

no spectra in Fig. 3 resembles “nqp” structure type. All spectra in the Fig. 3G-N show the same intensity of two bands around 1400 cm⁻¹. For comparison, in the Raman spectrum of “nqp” structure depicted in Supplementary Fig. 21 band at ≈1390 cm⁻¹ has twice the intensity of that at ≈1410 cm⁻¹. The variation between Figure 3G-J and K-N seems to result from differences between fully hydrated “lcp structure” (1·9/2H₂O) and evacuated “lcp” structure, with little effect to the coordination skeleton and no presence of 1·4/3H₂O. Again, this experiment demonstrated different water adsorption dynamics for “nqp” structure and “lcp” structure, but not the existence of “nqp” and “lcp” structure within a single crystal.

Response: According to your suggestion, we have re-analyzed the Raman spectrum in details and the sentence in the manuscript have been revised.

Lines 24-27 in page 6 and lines 1-7 in page 7 of the revised manuscript:

In order to further verify that the activation-method dependent adsorption properties are related to structural difference between nqp and partially dehydrated frameworks, micro-Raman spectra experiments (spot diameter ~2 μm) were performed on single crystals^{56,57}. As shown in Supplementary Fig. 19, the Raman spectrums of lcp and nqp phases present delicate difference in two regions: the bands around 1170 cm⁻¹ (C–C stretch modes of prentz ligands) and 1140 cm⁻¹ (CH-bending of prentz ligands)⁵⁸. In the nqp single crystal, the bands at around 1170 and 1140 cm⁻¹ persisted up to 3 days in saturated steam, and then covered to the characteristic bands of lcp structure type. For comparison, the bands at around 1140 cm⁻¹ remained unchanged as the lcp single crystal 1·9/2H₂O under mild activation (a constant N₂ gas), suggesting the lcp structural characteristics of partially dehydrated framework. The Raman spectrum of partially dehydrated single crystal can change to that of lcp structure type within 10 min in a humidity of 80%. The distinct responses of nqp and partially dehydrated single crystals in the micro-Raman spectra verify the structural dependent water adsorption property in this material.

Supplementary Figures 19 and 20 (caption) have also been modified.

Supplementary Fig. 19 Micro-Raman spectra of 4-amino-1,2,4-triazole, cinamyle aldehyde, prentz ligand, $K_2Pd(CN)_4$, lcp phase and nqp phase.

Fig. 20 Micro-Raman spectra of partially anhydrous (A) and nqp single crystals (B) before and after water adsorption.

9. One may argue that PXRD patterns combined with adsorption measurements seem to confirm phase-coexistence conclusion. However, as noted in point 3 of this review, the first diffraction peak is very misleading, as it is in the same position for **1** and $1\cdot4/3H_2O$. Please see for example Supplementary Fig. S15 – only two phases are accounted for: $1\cdot9/2H_2O$ “blue” and $1\cdot4/3H_2O$ “red”. Thus, the same “red” phase ($1\cdot4/3H_2O$) is assigned to patterns labelled as 360 min. and 2880 min., despite the vast difference in 15-30 degrees 2theta region. This makes analysis of this first diffraction peak in PXRD patterns very misleading and it may result in wrong conclusions about phase identity of the sample. I believe that PXRD pattern labelled as 2880 min. was obtained for the sample mostly composed of anhydrous **1**.

Response: In the previous Supplementary Fig. 15, the pattern labelled as 2880 min was tested after the sample cooling in air, which in fact rehydrated to $1\cdot4/3H_2O$. We have now replaced the PXRD pattern of anhydrous **1** cooling in air (rehydrated to $1\cdot4/3H_2O$) with that of anhydrous **1** cooling in vacuum (stable **1**) in Fig. 2 and Supplementary Fig. 15. The sentence “However, the resulting pure nqp phase could not recover rapidly to the lcp phase after being sprayed with water mist as in the former two cycles.” has been changed into “However, the resulting anhydrous **1** could rehydrate to nqp phase $1\cdot4/3H_2O$ within 60 min, while cannot recover to lcp phase smoothly upon water adsorption after being sprayed with water mist as in the former two cycles.” [lines 14-16, page 6].

The caption for **Fig. 2** has also been modified (see the response to question no. 4).

Supplementary Fig. 15 has also been modified, as shown below:

Supplementary Fig. 15 The full PXRD patterns during the water desorption-adsorption process. The sample was dehydrated rapidly upon heating at 393 K and the backward water adsorption was performed by spraying water mist on the sample. The partially dehydrated sample recovered to the lcp phase instantly, while the completely dehydrated sample 1 prepared from long-time heating (433 K) under vacuum can rehydrate to nqp phase 1·4/3H₂O within 60 min and cannot recover to lcp phase smoothly upon water adsorption. The all data were collected at room temperature after different activated treatment. The PXRD patterns of the physical mixture of two pure phases did not show any significant change after spraying the water mist for 60 min. The blue and red peaks denote the character peaks of structures of lcp and nqp phases, respectively.

10. "The SCO performances of phase-coexistence samples can be regarded as linear combinations of SCO transitions of 1·9/2H₂O and 1·4/3H₂O, and this phenomenon is significantly different from typical SCO compounds whose SCO behaviors are usually globally influenced by small changes in water content, suggesting the existence of interfaces between the lcp and nqp phases that interrupt the diffusion of lattice from lcp to nqp phase." [lines 214-221]

I strongly disagree with this conclusion. When properties of the mixture resemble linear combination of the ingredients, this means that there is no interaction between the ingredients. Physical mixture of 1·9/2H₂O and FeCl₂ will show linear combination of their magnetic properties, despite of the fact that 1·9/2H₂O and FeCl₂ do not show phase-coexistence within single crystals. It is well known that cooperative SCO (as observed for 1·9/2H₂O) is strongly affected by variation in crystallite size and similar interface effects. Thus existence of the interface between the lcp and the nqp is expected to affect magnetic properties of both phases, which is not observed in the data.

Response: In the previous description, the effect of the amount of water molecules on SCO behavior was ignored. Meanwhile, the SCO variation of partially dehydrated samples could not be ruled out as the linear combination of dehydrated and non-dehydrated particles. The magnetic description of partially dehydrated samples has now been revised and the magnetic graph has been split into Fig. 3 and Supplementary Fig. 22.

Changes made in lines 12-19 of page 8:

For the SCO properties of partially dehydrated samples, a hysteretic spin transition at 133–147 K emerged when $1.9/2\text{H}_2\text{O}$ was partially desolvated. Upon further water molecules losing, a series of variations in the total SCO properties were accompanied by the intermediate hysteresis loop increasing, resulting in the three- and four-step SCO. Though the SCO variation of partially dehydrated samples cannot be ruled out as the linear combination of dehydrated and non-dehydrated particles, the partially dehydrated samples with relatively minimal water molecules showed the incomplete SCO with the similar $T_{1/2}$ as that of nqp phase $1.4/3\text{H}_2\text{O}$, suggesting that the host-guest interaction affecting SCO behavior is determined by the amount of water molecules and dynamic framework⁶⁰⁻⁶².

Fig. 3 Temperature dependence of the $\chi_M T$ curves of lcp phase $1.9/2\text{H}_2\text{O}$ and nqp phase $1.4/3\text{H}_2\text{O}$ upon heating and cooling. The two-step SCO with a large plateau of lcp phase $1.9/2\text{H}_2\text{O}$ (A) and a two-step SCO without plateau of nqp phase $1.4/3\text{H}_2\text{O}$ (B).

Supplementary Fig. 22 Temperature dependence of the $\chi_M T$ curves of partially dehydrated samples. The SCO of the partially dehydrated samples exhibited a multi-step spin transition in which the degree of magnetic switching can be adjusted by controlling the water molecules losing (A–D). The corresponding state of each sample was characterized by PXRD.

In conclusion, I believe that the resubmitted manuscript contains mostly valid measurements (unfortunately

PXRD patterns suffer from strong texture, but that results from the experimental setup) and the observed phenomenon of “programmable” sorption is interesting, with the “lcp” structure showing twice the water uptake of the “nqp” structure. However, analysis of the resubmitted version led me to the conclusion, that this behavior can be fully explained by slow lcp-nqp dynamics. In my opinion there is no evidence for phase-coexistence within the single crystals of this material. Therefore I encourage the authors to re-analyze relevant measurements and prepare a revised text, with emphasis on the interconversion between anhydrous 1, “nqp” structure type and “lcp” structure type”, rather than questionable phase-coexistence phenomenon (in my opinion no additional measurements are necessary to fully describe behavior of this framework in the absence of phase-coexistence hypothesis).

Response: Thank you for the recognition of innovation in this work and the valuable suggestions concerning the manuscript. We have re-analyzed relevant measurements and prepare a revised text according to your suggestion. The main changes are as follows:

- 1) The title of manuscript “Single crystal-to-single crystal and spin-crossover transformations via phase-coexistence state in pore-adjustable frameworks” has been changed into “A spin-crossover framework endowed with pore-adjustable behavior by slow structural dynamics”.
- 2) The “phase-coexistence state” has been changed into “partially dehydrated samples” or “partially dehydrated framework”.
- 3) The programmable sorption has been explained as “slow lcp-nqp dynamics” and “a pore rearrangement associated with simultaneous pore opening and closing”.
- 4) Figure out the conversion condition between anhydrous 1, nqp phase $1.4/3H_2O$ and lcp phase $1.9/2H_2O$. Anhydrous 1 and nqp phase $1.4/3H_2O$ are not exactly the same phase.

The Conclusion section has also been modified, as shown below marked in yellow: The simultaneous pore opening and closing involved in the pore rearrangement leads to slow nqp-lcp dynamics, where nqp phase need to long-time exposure in saturated steam to accomplish the nqp-to-lcp gate-opening adsorption. Moreover, the structural transformation of the magnetic framework shifts the SCO properties of the Fe^{II} magnetic centers. This study presents an unprecedented adsorption-related pore transformation accompanied by activation-method dependent adsorption. Such an exotic property can be used in intelligently adjustable gas adsorption, actuation, and sensing. [lines 1-6, Page 11]

In the section of Introduction, the sentence “Therefore, the intercorrelation between the guests and SCO frameworks potentially leads to intriguing magnetic switching that directly entangles the local and global structural transformations involved in guest adsorption.” has been changed into “Therefore, the intercorrelation between the guests and SCO frameworks potentially leads to intriguing magnetic switching that directly entangles the structural transformations involved in guest adsorption. [Page 2, line 26; Page 3, lines 1-2]. While the sentences “The atypical pore transformation is accompanied by a distinct adsorption process in which a stable phase-coexistence state, where narrow quasi-discrete pore (nqp) and large channel-type pore (lcp) phases are concomitant in an individual crystal, needs to be activated first by long-time exposure in saturated steam to accomplish the nqp-to-lcp gate-opening transition. A direct guest-adjustable multistep spin transition is realized as a reflection of phase coexistence because of the intercorrelations between the flexible framework and the SCO properties.” have been changed into “The atypical pore transformation is accompanied by a distinct adsorption process in which to be activated first by long-time exposure in saturated steam to accomplish the slow dynamics between nqp and lcp structures. The direct variant multistep spin transition of the nqp and lcp phases is as a reflection of the flexible framework and the host-guest interactions.” [lines 8-12, Page 3].

Additional comments:

11. “the decomposition of framework of 1 should be precluded, as we can re-obtained the crystal structure of 1·4/3H₂O by exposing one of the above single crystals in the air” [rebuttal, page 11]. The ability to re-obtain the crystalline form by re-solvation does not confirm the stability of the parent framework. Recently there were several reports regarding solvent-driven transformation from amorphous to crystalline state (Dalton Trans., 2018, 47, 845; J. Am. Chem. Soc. 2021, 143, 20202 and for cyanides: Chem. Sci., 2021, 12, 9176).

Response: The sentences “Notably, the PXRD pattern after the measurement was same with that before adsorption experiments, suggesting the robust crystal lattice of this sample (Supplementary Fig. 11)” have been changed into “Notably, the PXRD pattern after two isothermal vapor ad/de-sorption or the adsorption-desorption-cycles measurement revealed that the crystallinity and framework of nqp and lcp structure type samples can both be revived by rehydration (Supplementary Fig. 11).” [line 26 in Page 5 and lines 1-2 in Page 6].

12. Supporting Fig. 11 – it would be more appropriate to compare the sample after sorption experiments with the already activated sample (Supp. Fig. S11A – I believe that PXRD pattern from Supp. Fig. S8 may be used for that).

Response: Supplementary Fig. 11 has been modified according to your suggestion:

Supplementary Fig. 11 PXRD patterns of samples after in situ activation (Blue) and after two adsorption isotherm experiments and the continuous adsorption-desorption-cycles measurement (Green). A, The PXRD pattern of sample after two isothermal vapor ad/de-sorption (in the Fig. 2A up) of the sample activated at 433 K under vacuum for 48 h. B, The PXRD pattern of sample after a continuous adsorption-desorption-cycles measurement (in the Fig. 2B, including a long-time heating at 433 K under vacuum and a mild condition at 303 K under vacuum) reveals that the crystallinity and framework of this sample can be revived by rehydration.

13. On several occasions experimental patterns were compared to the simulated pattern for 1.4/3H₂O at 100 K (for example Supp. Fig. S13) – please replace it with the simulated pattern for structure at 250 K.

Response: The simulated pattern for 1·4/3H₂O has been replaced with that for structure at 250 K in Supplementary Fig. 8, 12–14.

14. Fig. 2B – “activation” arrow should end around 1300 minutes.

Response: We know that there are two parts of the activated process, i.e. 300 K under vacuum and heating

under vacuum. The time of high temperature activation and adsorption have been re-checked and modified.

Lines 15-19 of Page 5 in the revised Manuscript:

As shown in Fig. 2B, when the sample prepared from a long-time heating (14 h) at 433 K under vacuum condition was placed in a steam atmosphere with $P/P_0 = 0.9$, it underwent two-step water adsorption: a rapid partially water adsorption of $\sim 1.7 \text{ H}_2\text{O}$ per Fe^{II} , and later a very slow water adsorption (97 h) of $\sim 4.5 \text{ H}_2\text{O}$ per Fe^{II} that reflected the $1.4/3\text{H}_2\text{O}$ -to- $1.9/2\text{H}_2\text{O}$ gate-opening structural change.

The caption for Supplementary Fig. 10 has also been modified:

Supplementary Fig. 10 The water adsorption-desorption measurements of the sample activated at different temperatures. (A), The sample activated at 400 K under vacuum for 22 h. (B), The sample activated at 433 K under vacuum for 14 h. Although the measurements indicate that almost all water guests are removed upon heating at 400 K, the uptake of ca. $2.6 \text{ H}_2\text{O}$ per Fe^{II} for the sample activated at 400 K for 22 h is larger than that of ca. $1.7 \text{ H}_2\text{O}$ per Fe^{II} for the sample activated at 433 K for 14 h, suggesting more crystals of partially dehydrated framework survived in the former, and this result is consistent with the PXRD measurements.

15. Article, page 14 – in the description of Raman setup millimeters should probably be replaced with micrometers.

Response: The Raman setup in this work was also used in elsewhere (Chemosphere, 2020, 241, 124960). We have re-checked and modified the parameters of the Raman setup as blow: Laser beam was focused on the selected region of single crystal through a 50 × objective lens (0.75 numerical aperture), which provided spatial resolution of about $6 \mu\text{m}$. [lines 22-23, page 13]

Response to reviewer #2:

Comments to Authors:

Many of reviewer 1's comments on the original manuscript, and some of mine, concerned the crystallography. That has been addressed by remeasuring or re-refining all the structures in the study, including some new data collections with synchrotron radiation. The new structural data are a significant improvement on the originals, and the many of the experimental issues itemised by reviewer 1 have now been addressed. The new crystal structures from the nqp phase are of reasonable quality, for the product of a single-crystal-to-single-crystal annealing reaction.

The revision also includes new thermogravimetric analyses and extra magnetic data (which I requested), and single crystal Raman microscopy (to address a comment by reviewer 1). Those are all helpful additions.

I have just a few new comments to address before final acceptance of the manuscript.

Response: *We thank the reviewer for positive comments on our work and valuable suggestions concerning the manuscript.*

1. My only criticism of the new data, is that the Raman microscopy data in Supplementary Figure 23 don't clearly show the co-existence of both phases in the same crystal, as proposed in the caption. Spectra E-H all look the same; I-L all look the same; and M-P all look the same. So, the phase composition at all four points of the crystal looks the same, under each of the conditions measured. So, that experiment was worth doing, but it doesn't prove the two phases can co-exist in the same crystal, at the same time. At best, what it shows is that the domains of the two phases in the mixed-phase crystal must be smaller than the resolution of the Raman microscope. The caption to Supplementary Figure 23, and the description on lines 167-175, should be changed to reflect that. Supplementary Figure 19 does show the two phases can co-exist in the same crystal, though, so that conclusion is sound on a macroscopic level, even if it was not observed microscopically.

Response: *According to your suggestion, we have re-analyzed the Raman spectrum in details and the related sentences in the manuscript have been revised.*

Lines 24-27 in page 6 and lines 1-7 in page 7: In order to further verify that the activation-method dependent adsorption properties are related to structural difference between nqp and partially dehydrated frameworks, micro-Raman spectra experiments (spot diameter $\sim 2 \mu\text{m}$) were performed on single crystals^{56,57}. As shown in Supplementary Fig. 19, the Raman spectrums of lcp and nqp phases present delicate difference in two regions: the bands around 1170 cm^{-1} (C-C stretch modes of prentz ligands) and 1140 cm^{-1} (CH-bending of prentz ligands)⁵⁸. In the nqp single crystal, the bands at around 1170 and 1140 cm^{-1} persisted up to 3 days in saturated steam, and then converted to the characteristic bands of lcp structure type. For comparison, the bands at around 1140 cm^{-1} remained unchanged as the lcp single crystal $1.9/2\text{H}_2\text{O}$ under mild activation (a constant N_2 gas), suggesting the lcp structural characteristics of partially dehydrated framework. The Raman spectrum of partially dehydrated single crystal can change to that of lcp structure type within 10 min in a humidity of 80%. The distinct responses of nqp and partially dehydrated single crystals in the micro-Raman spectra verify the structural dependent water adsorption property in this material.

In addition, Supplementary Fig. 19 and 20 (caption) have also been modified.

Supplementary Fig. 19 Micro-Raman spectra of 4-amino-1,2,4-triazole, cinamyle aldehyde, prenrz ligand, $\text{K}_2\text{Pd}(\text{CN})_4$, lcp phase and nqp phase.

Supplementary Fig. 20 Micro-Raman spectra of partially anhydrous (A) and nqp single crystals (B) before

and after water adsorption.

2. Line 153. The abbreviation 'SCP' is not in common use, and isn't defined anywhere. The 'CP' is presumably Coordinated Polymer, but I don't know what the 'S' stands for. Since it isn't used anywhere else in the manuscript, that abbreviation should be changed for something more obvious.

Response: *The sentence containing this abbreviation has been deleted from the manuscript.*

3. An English error that comes up several times is "pore open and close", which should be changed to "pore opening and closing". More generally, while the manuscript is always understandable, there are many small English mistakes and the text would benefit from English corrections at the proof stage.

Response: *Many thanks for your kind suggestion. we have checked the manuscript and corrected some English mistakes, including the change of "pore open and close" into "pore opening and closing".*

Reviewer #1 (Remarks to the Author):

In my opinion, the revised manuscript contains satisfactory description of the most important scientific finding – the dependence of water sorption on the activation method. This is supported by the detailed characterization of the material with use of numerous physical measurements, which are presented in an appropriate manner, allowing the reader to independently evaluate of the results. In this form, I believe that the article deserves publication.

I have 3 minor comments regarding the final version of the manuscript, but they mainly relate to the wording and probably can be addressed at the article production stage:

1) In the caption to Figure 2A: “physical mixture (1·9/2H₂O and 1·4/3H₂O)” could be substituted with “physical mixture (nqp and lcp phases)” or similar. Physical mixture of 1·9/2H₂O and 1·4/3H₂O exists mostly at $p/p_0 = 0.9$, at $p/p_0 = 0$ it is rather a mixture of 1·9/2H₂O and anhydrous 1, so using lcp/nqp names seems more appropriate.

2) Supplementary Fig. 11A shows that PXRD of the sample after in situ activation at 433 K is not the same as the one after ad/de-sorption experiment. This is in line with the former being anhydrous 1 and the latter being nqp phase. This may not be obvious to the reader and deserves a short comment in the text of the main article, where Supp. Fig. 11 is mentioned.

3) In the Supplementary Fig. 20 “partially anhydrous single crystal” should rather be “partially dehydrated single crystal”, in line with other uses of “partially dehydrated” throughout the article.

Response to reviewer #1:

Comments to Authors:

In my opinion, the revised manuscript contains satisfactory description of the most important scientific finding – the dependence of water sorption on the activation method. This is supported by the detailed characterization of the material with use of numerous physical measurements, which are presented in an appropriate manner, allowing the reader to independently evaluate of the results. In this form, I believe that the article deserves publication.

Response: We would like to thank all reviewers once again for their insightful comments.

I have 3 minor comments regarding the final version of the manuscript, but they mainly relate to the wording and probably can be addressed at the article production stage:

1) In the caption to Figure 2A: “physical mixture (1·9/2H₂O and 1·4/3H₂O)” could be substituted with “physical mixture (nqp and lcp phases)” or similar. Physical mixture of 1·9/2H₂O and 1·4/3H₂O exists mostly at $P/P_0 = 0.9$, at $P/P_0 = 0$ it is rather a mixture of 1·9/2H₂O and anhydrous 1, so using lcp/nqp names seems more appropriate.

Response: In the caption of Figure 2A, the “physical mixture (1·9/2H₂O and 1·4/3H₂O)” has been replaced with “physical mixture (nqp and lcp phases)”.

2) Supplementary Fig. 11A shows that PXRD of the sample after in situ activation at 433 K is not the same as the one after ad/de-sorption experiment. This is in line with the former being anhydrous 1 and the latter being nqp phase. This may not be obvious to the reader and deserves a short comment in the text of the main article, where Supp. Fig. 11 is mentioned.

Response: A short comment “Notably, the PXRD pattern after two isothermal vapor ad/de-sorption or the adsorption-desorption-cycles measurement revealed that the crystallinity and framework of nqp and lcp structure type samples can both be revived by rehydration, and further reflected the different structural flexibilities between anhydrous 1, nqp phase 1·4/3H₂O

and lcp phase $1.9/2H_2O$ (Supplementary Fig. 11).” has been added in revised Manuscript. (lines 1-2, Page 6)

3) In the Supplementary Fig. 20 “partially anhydrous single crystal” should rather be “partially dehydrated single crystal”, in line with other uses of “partially dehydrated” throughout the article.

Response: The “partially anhydrous single crystal” in the Supplementary Fig. 20 has been replaced with “partially dehydrated single crystal”.